# Mechanisms of site-specific dephosphorylation and kinase opposition imposed by PP2A regulatory subunits

Thomas Kruse[1,†] (iD), Sebastian Peter Gnosa[2,†], Isha Nasa[3,4,†], Dimitriya Hristoforova Garvanska[1,†],
Jamin B Hein[1], Hieu Nguyen[3,4], Jacob Samsøe-Petersen[2], Blanca Lopez-Mendez[1],
Emil Peter Thrane Hertz[1], Jeanette Schwarz[2], Hanna Sofia Pena[2], Denise Nikodemus[2] (iD),
Marie Kveiborg[2,*] (iD), Arminja N Kettenbach[3,4,**] (iD) & Jakob Nilsson[1,***] (iD)

## Abstract

PP2A is an essential protein phosphatase that regulates most cellular processes through the formation of holoenzymes containing distinct regulatory B-subunits. Only a limited number of PP2A-regulated phosphorylation sites are known. This hampers our understanding of the mechanisms of site-specific dephosphorylation and of its tumor suppressor functions. Here, we develop phosphoproteomic strategies for global substrate identification of PP2A-B56 and PP2A-B55 holoenzymes. Strikingly, we find that B-subunits directly affect the dephosphorylation site preference of the PP2A catalytic subunit, resulting in unique patterns of kinase opposition. For PP2A-B56, these patterns are further modulated by affinity and position of B56 binding motifs. Our screens identify phosphorylation sites in the cancer target ADAM17 that are regulated through a conserved B56 binding site. Binding of PP2A-B56 to ADAM17 protease decreases growth factor signaling and tumor development in mice. This work provides a roadmap for the identification of phosphatase substrates and reveals unexpected mechanisms governing PP2A dephosphorylation site specificity and tumor suppressor function.

**Keywords** ADAM17; phosphoproteomics; PP2A; substrate specificity; tumor suppressor
**Subject Categories** Cell Cycle; Post-translational Modifications & Proteolysis; Proteomics
The EMBO Journal (2020) 39: e103695

## Introduction

It has been appreciated for over 40 years that multisite phosphorylation is a major mechanism for regulating protein function (Cohen, 2000). Historically, the differential phosphorylation of a protein was attributed to the actions of tightly regulated kinase activities. In this scenario, kinases are the major determinants of multisite phosphorylation with protein phosphatases being promiscuous counteractors. Kinases achieve specificity through a deep catalytic cleft that recognize specific amino acid consensus sequences flanking the phosphorylation site as well as through interactions with short linear motifs (SLiMs) in substrates (Miller & Turk, 2018). Our understanding of kinase function has been greatly facilitated by combining specific inhibitors with phosphoproteomics allowing global substrate identification. However, our understanding of phosphatase substrates is currently limited due to lack of specific inhibitors.

Phosphoprotein phosphatase 2A (PP2A) accounts for the majority of phosphoserine and phosphothreonine dephosphorylation in eukaryotic cells and regulates many aspects of cellular physiology (Virshup & Shenolikar, 2009; Nilsson, 2019). PP2A is a heterotrimer composed of a catalytic C-subunit, a scaffolding A-subunit, and a regulatory B-subunit. The B-type subunits belong to four distinct gene families, B (B55), B′ (B56), B″ (PR72), and B‴ (PR93), each encoding two to five isoforms (Virshup & Shenolikar, 2009; Wlodarchak & Xing, 2016). The B55 and B56 families are the largest of the regulatory subunit families comprising four and five human isoforms, respectively.

Recent discoveries have shown that PP2A similarly to kinases interacts with substrates and substrate specifying proteins through SLiMs (Cundell et al, 2016; Hertz et al, 2016; Wang et al, 2016; Wu et al, 2017). In the case of PP2A-B56, a series of biochemical and

1 Novo Nordisk Foundation Center for Protein Research, Faculty of Health and Medical Sciences, University of Copenhagen, Copenhagen, Denmark
2 Biotech Research and Innovation Centre (BRIC), University of Copenhagen, Copenhagen, Denmark
3 Biochemistry and Cell Biology, Geisel School of Medicine at Dartmouth College, Hanover, NH, USA
4 Norris Cotton Cancer Center, Lebanon, NH, USA
  *Corresponding author. Tel: +45 35325679; E-mail: marie.kveiborg@bric.ku.dk
  **Corresponding author. Tel: +1 603 653 9068; E-mail: Arminja.N.Kettenbach@dartmouth.edu
  ***Corresponding author. Tel: +45 21328025; E-mail: jakob.nilsson@cpr.ku.dk
  †These authors contributed equally to this work

structural studies have revealed that proteins containing LxxIxE type of motifs engage a conserved binding pocket on all isoforms of B56 regulatory subunits (Hertz *et al*, 2016; Wang *et al*, 2016). For PP2A-B55, clusters of basic residues surrounding the phosphorylation site likely bind to a conserved acidic surface on the B55 subunit (Cundell *et al*, 2016). However, once bound to their substrates to which extent and how the PP2A holoenzymes selectively dephosphorylate individual sites is not known. One possibility is that substrate binding *per se* provides the proper three-dimensional positioning (key-in-lock model) of the PP2A active site for some phosphorylation sites but not others (Xu *et al*, 2006, 2008; Cho & Xu, 2007). Alternatively, it has been suggested that, like kinases, phosphatases may favor certain amino acid sequences immediately surrounding the phosphorylation site (Ubersax & Ferrell, 2007; Saraf *et al*, 2010; McCloy *et al*, 2015). However, only a small number of phosphorylation sites have been experimentally linked to specific PP2A holoenzymes making it difficult to conclude on general principles of phosphorylation site specificity. Furthermore, a direct comparison of substrates for two PP2A holoenzymes would be needed to determine whether regulatory subunits only act as targeting subunits or have additional roles in site-specific dephosphorylation. Uncovering this would have important implications for understanding how phosphorylation-mediated signaling is regulated.

## Results

### Development of a specific PP2A-B56 inhibitor

To understand principles of PP2A specificity, we focused on PP2A-B56 which is a major tumor suppressor (Janssens *et al*, 2005; Eichhorn *et al*, 2009). Proteins containing LxxIxE motifs engage a conserved binding pocket on B56 regulatory subunits with varying micromolar affinities depending on the exact amino acid composition of the motif (Hertz *et al*, 2016; Wang *et al*, 2016; Wu *et al*, 2017). We recently showed that high-affinity LxxIxE motifs, when expressed *in vivo*, inhibit dephosphorylation by PP2A-B56 of the Ebola VP30 transcription factor (Kruse *et al*, 2018), probably by acting like a competitive inhibitor displacing PP2A-B56 from its substrate. Provided sufficient specificity and potency, we reasoned that such an inhibitor could be used to displace PP2A-B56 from all cellular LxxIxE containing interactors and, thus, to interrogate the phosphoproteome regulated by this phosphatase. To test this, we designed a series of constructs containing 1, 2, or 4 copies of a functional high-affinity LxxIxE motif separated by spacer sequences and fused these to either polyhistidine (His-tag) or yellow fluorescent protein (YFP) (Fig 1A). Constructs containing 4 copies of a nonbinding AxxAxA motif were included as controls. The His-tagged inhibitor series were expressed and purified from *Escherichia coli*, and their binding to recombinant B56α was analyzed using isothermal titration calorimetry (ITC). Indeed, both the binding affinities ($K_D$) and the stoichiometry (number of B56 molecules bound per inhibitor) increased with the number of LxxIxE motifs (Figs 1A and B, and EV1A). Prolonged mitosis is a well-established mitotic phenotype of interfering with PP2A-B56 function (Foley *et al*, 2011; Suijkerbuijk *et al*, 2012; Kruse *et al*, 2013). To assess the potency of the inhibitor series in cells, the YFP-tagged constructs were transfected into HeLa cells and progression through mitosis was

monitored by live-cell microscopy. A clear correlation between phenotype severity and inhibitor copy number was observed (Fig 1C). Based on these results, we focused on the YFP-tagged 4x (LxxIxE) B56 inhibitor and the corresponding 4x(AxxAxA) control inhibitor. To determine the specificity of the B56 inhibitor, it was affinity-purified from HeLa cells and interacting proteins were identified by quantitative label-free mass spectrometry (MS). Strikingly, all components of the PP2A-B56 holoenzyme were strongly enriched in elutes from B56 inhibitor samples compared to control inhibitor samples (Fig 1D and Table EV1). This includes the five isoforms of B56 regulatory subunits and the two isoforms of each of the catalytic and scaffold subunits. NSF (N-ethylmaleimide-sensitive fusion protein), a vesicle-fusing ATPase, was the only other protein that bound specifically to the B56 inhibitor. Thus, the B56 inhibitor displays excellent specificity toward the PP2A-B56 holoenzyme family. We also concluded from this experiment that most proteins directly interacting with PP2A-B56 engage the LxxIxE binding pocket for effective binding.

Next, we tested whether the B56 inhibitor is able to displace PP2A-B56 interactors. To this end, YFP-B56α was transfected into HeLa cells stably expressing mCherry-tagged B56 inhibitor or control inhibitor. Purifications of YFP-B56α were subsequently analyzed by quantitative label-free MS (Fig 1E and Table EV2) or Western blotting (WB) (Fig 1F) probing with antibodies against Separase, Kif4A, BubR1, and Axin1, which all contain validated LxxIxE motifs (Hertz *et al*, 2016). Both MS and WB analyses revealed that YFP-B56α enriched from cells expressing the B56 inhibitor shows a significant decrease in interactor binding compared to YFP-B56α enriched from cells expressing the control inhibitor.

In summary, we have developed a competitive inhibitor with excellent potency and specificity toward PP2A-B56.

### Identification of PP2A-dependent phosphorylation sites

We next used this inhibitor to identify *in vivo* substrates of PP2A-B56. Stable HeLa cell lines allowing rapid induction of the B56 or control inhibitor were synchronized at either G1/S or in mitosis (M), and cells were collected and processed for quantitative phosphoproteomics analysis (Figs 2A and B, and EV2A and B). Using this approach, we identified and quantified a total of 13,515 and 27,745 phosphorylation sites in G1/S and M, respectively (Fig 2A and B, Table EV3). Of these sites, 548 and 398 were significantly increased in phosphorylation upon B56 versus control inhibitor expression ($\log_2$ ratio > 0.8 (1.75-fold), *P*-value < 0.05, phosphorylation site localization probability > 75%) in G1/S and M, respectively. The phosphorylation sites that increased were located on 651 proteins of which 34 contain a validated or predicted LxxIxE motif (Hertz *et al*, 2016; Wu *et al*, 2017). It was previously shown that LxxIxE containing proteins can act as scaffolds for the recruitment of other proteins for dephosphorylation (Suijkerbuijk *et al*, 2012; Qian *et al*, 2017; Kruse *et al*, 2018). Using the STRING database (Jensen *et al*, 2009), we found that an additional 491 of the 651 proteins identified in our screen are direct interactors of proteins with validated or predicted LxxIxE docking motifs. Thus, the majority of up-regulated phosphorylation sites are present on LxxIxE containing proteins or on their immediate interactors strongly supporting the notion that the identified phosphorylation sites are PP2A-B56 targets.

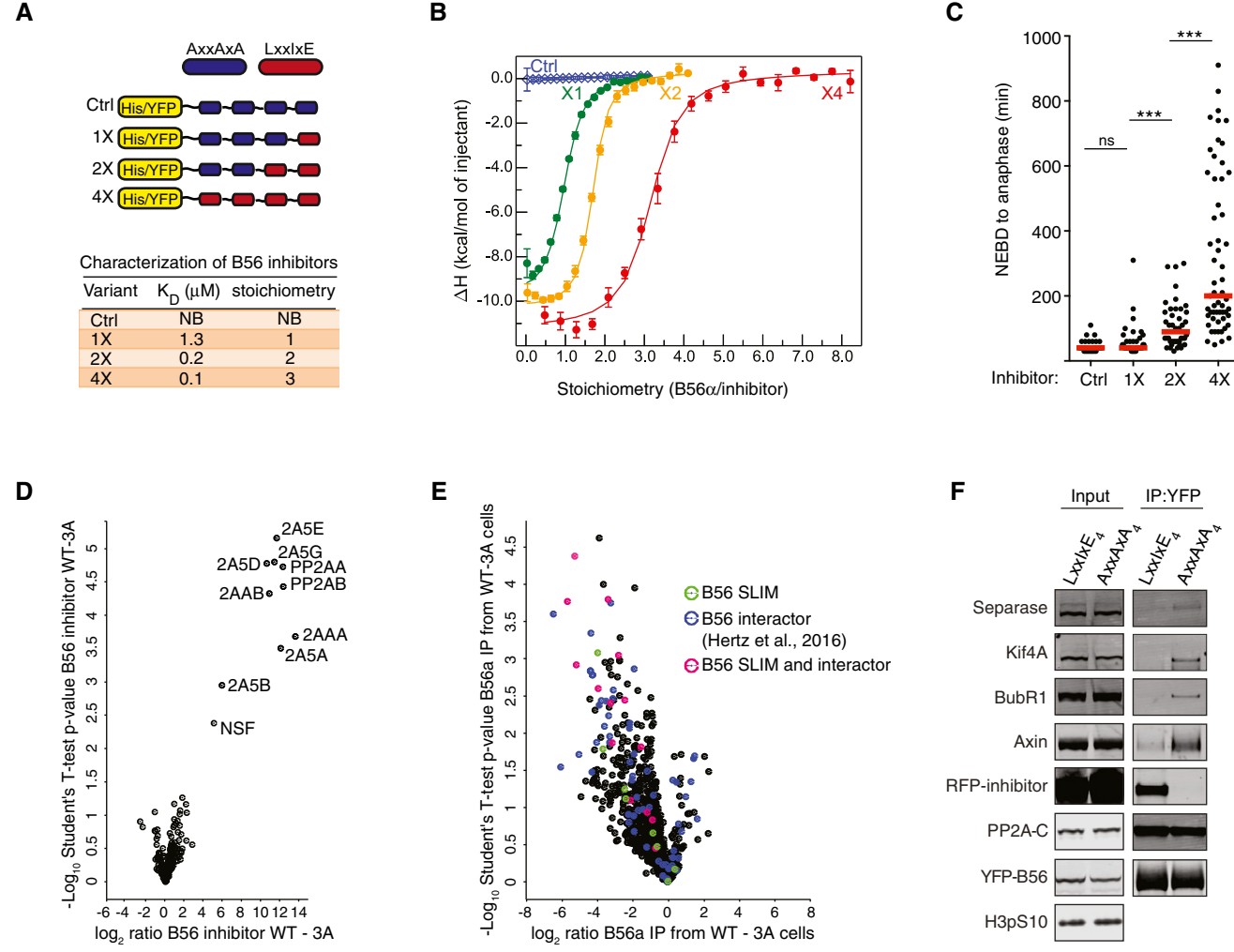

**Figure 1. Development of a PP2A-B56 specific inhibitor.**

A, B  Schematic of the B56 inhibitor series and affinities and stoichiometry's for B56α measured by ITC. Global direct fitting shown for one experiment (reverse). Each dot is the integrated heat per injection, and the error bars represent uncertainty with this integrated value. The experiment was done in both direct (B56 in cell) and reverse (B56 in syringe) with similar results.

C  Time from nuclear envelope breakdown (NEBD) to mitotic exit of cells expressing the indicated B56 inhibitors with each circle representing a single cell. Only cells with similar expression levels of the various B56 inhibitor constructs were analyzed. Median time is indicated by red line. A representative result from at least three independent experiments is shown. At least 25 cells were counted per condition in the experiment shown. A Mann–Whitney *U*-test was used for statistical analysis (ns: non-significant, \*\*\**P* ≤ 0.001).

D  Volcano plot representing mass spectrometry identified proteins co-purifying with B56 inhibitor versus control inhibitor from HeLa cells. PP2A-B56 subunits co-purifying with the B56 inhibitor are indicated.

E, F  Competition assay in HeLa cells stably expressing RFP-tagged B56 inhibitor (LxxIxE) or control inhibitor (AxxAxA). YFP-B56α was transfected into and subsequently purified from these cell lines. Loss of binding of indicated proteins determined by either mass spectrometry (pink—B56 SLiM-containing protein and known B56 interactor; blue—known B56 interactor, green—B56 SLiM-containing protein) (E) or Western blotting (F).

In general, dephosphorylation of a site was a specific event that did not affect all phosphorylation sites on a protein. Comparison of $\log_2$ ratios of B56-dependent dephosphorylation sites versus other phosphorylation sites on the same protein was not correlated ($R = 0.1113$; Fig EV2C). To explore the mechanism for this differential site specificity, the chemical nature of the phosphorylation sites dephosphorylated by PP2A-B56 was investigated. Over- and under-representation of amino acids surrounding the up-regulated phosphorylation site were determined by comparison with all phosphorylation sites identified in the respective screens (Colaert *et al*, 2009). This revealed a preference for basic amino acids upstream of

the dephosphorylation site (Fig 2C and D). Interestingly, we found a strong deselection for phosphorylation sites containing a proline residue in the +1 position. In contrast, no preference pattern was observed when we analyzed the 184 and 163 phosphorylation sites that were significantly decreased in phosphorylation upon B56 versus control inhibitor expression ($\log_2$ ratio > 0.8 (1.75-fold), *P*-value < 0.05, phosphorylation site localization probability > 75%) in G1/S and M, respectively (Fig EV2D). Furthermore, among all increased phosphorylation sites co-identified in both G1/S and M datasets, the overlap of PP2A-B56-regulated sites was only 4% (Fig EV2E). This indicates unique B56 substrates in mitosis and G1/

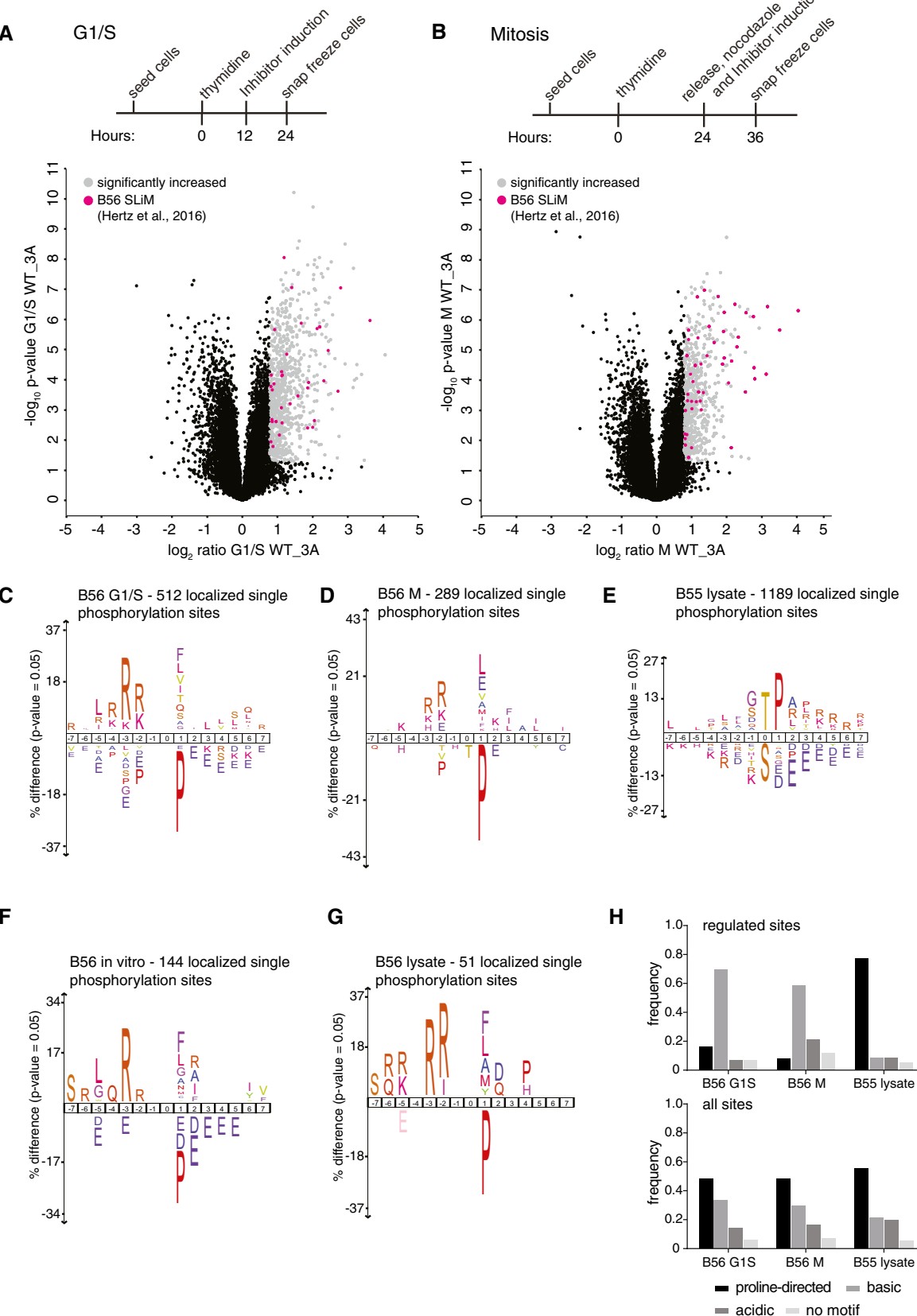

Figure 2.

**Figure 2. Phosphorylation site preference of PP2A-B56.**

A, B   Schematic of synchronization protocol for G1/S (A) or mitotic (B) arrested cells and accompanying volcano plot of phosphorylation sites quantified. The phosphorylation sites showing an increase in the presence of the B56 inhibitor are shown in light gray. Phosphorylation sites in a protein containing an LxxIxE motif are colored pink.

C–E   IceLogo representation of over- and underrepresented amino acid residues surrounding phosphorylation sites for the indicated experiments (letter coloring is standard iceLogo color output).

F, G   IceLogo representation of over- and underrepresented amino acid residues surrounding phosphorylation sites for the experiments indicated in Fig EV2F and G, respectively (letter coloring is standard iceLogo color output).

H   Distribution of phosphorylation site consensus motifs surrounding B56- or B55-dependent up-regulated phosphorylation sites (top panel) in comparison with all phosphorylation sites (bottom panel) in the indicated experiment (proline-directed: pS/TP; basic: R/KxxpS/T, R/KxpS/T, R/KpS/T; acidic: D/E/NxpS/T, D/EpS/T, pS/TD/E, pS/TxD/E; x—any amino acid).

S, yet the datasets reveal remarkably similar phosphorylation site consensus motif preferences.

The phosphorylation site preference of PP2A-B56 was further investigated by two independent experimental methods. First, we performed an *in vitro* peptide phosphatase assay that sampled phosphopeptides purified from cells. Purified PP2A-B56α holoenzyme was added and dephosphorylation kinetics followed by mass spectrometry (Figs 2F and EV2F, Table EV4). Second, we added an LxxIxE inhibitor peptide or a control peptide to cell extracts and determined inhibition of dephosphorylation after 5 and 15 min (Figs 2G and EV2G, Table EV5). Both approaches confirmed the apparent preference for basophilic and a deselection of proline-directed phosphorylation sites. Moreover, neither of the PP2A-B56 phosphoproteomics screens revealed a preference of this phosphatase for phosphorylated threonines over phosphorylated serines.

These findings are in stark contrast to previous observations that proline-directed phosphorylation sites are excellent substrates of PP2A-B55, another major PP2A holoenzyme species, and that this phosphatase shows a clear preference for phosphothreonine over phosphoserine (Agostinis *et al*, 1992; Cundell *et al*, 2016; Godfrey *et al*, 2017). To investigate this difference further, we identified PP2A-B55-regulated phosphorylation sites by adding thiophosphorylated Arpp19, a potent mitotic PP2A-B55 inhibitor (Gharbi-Ayachi *et al*, 2010; Mochida *et al*, 2010), or thiophosphorylated Arpp19 S62A as a control to mitotic cell extracts and compared these samples quantitatively by mass spectrometry or Western blotting with α-pTP antibodies (Fig EV2H). This identified 1405 PP2A-B55 up-regulated sites ($\log_2$ ratio > 0.8 (1.75-fold), *P*-value < 0.05, phosphorylation site localization probability > 75%) of which less than 1.3% was shared with the sites identified in the PP2A-B56 datasets. Moreover, these results confirmed the differential preference of PP2A-B55 and PP2A-B56 for proline-directed motifs and the previously reported preference of PP2A-B55 for phosphothreonine over phosphoserine (Fig 2E and Table EV6). Comparison of all increased phosphorylation sites in the PP2A-B56 G1/S and M as well as the PP2A-B55 datasets supports the notion that while basophilic, acidophilic, and proline-directed sites can be dephosphorylated by the two phosphatase holoenzymes, PP2A-B56 and PP2A-B55 show a clear difference in their relative preferences (Fig 2H).

### Differential phosphorylation site preference of PP2A-B56 and PP2A-B55

Next, we wanted to investigate whether the observed dephosphorylation site patterns are inherent properties of the two PP2A holoenzymes. In principle, preference of certain phosphorylation sites as observed in the phosphoproteomics analysis could be biased by

PP2A holoenzymes localizing to specific subcellular compartments enriched for certain kinases. First, we established an *in vitro* phosphatase assay with purified PP2A-B56α and PP2A-B55α holoenzymes using synthetic phosphopeptides with phosphorylation sites surrounded by amino acids conforming to physiologically relevant basophilic (PKC), acidophilic (PLK1), or proline-directed (CDK1) kinase consensus sequences (Fig 3A). Kinetic analysis revealed that the Michaelis–Menten constant $K_m$ was roughly similar for PP2A-B56 and PP2A-B55 toward the three different phosphopeptides. On the other hand, the catalytic efficiency ($K_{cat}/K_m$) of PP2A-B56 toward the proline-directed phosphopeptide was 50- to 100-fold lower compared to the acidophilic and basophilic ones, whereas this difference was not observed for the PP2A-B55 holoenzyme.

The PP2A-B55 phosphoproteomics experiment (Fig 2E) predicts a preference for basic amino acids and a deselection of acidic residues C-terminal to the phosphorylated TP sites. Deselection of basic amino acid residues was observed N-terminal to the TP sites. A panel of synthetic peptides with a variable content of amino acid residues flanking the phosphorylated TP site almost completely recapitulated this *in vitro* (Fig 3B). The PP2A-B56 iceLogos predict a preference for basic amino acids N-terminal to the phosphorylation site and a deselection of prolines in the +1 position, whereas no phosphothreonine over phosphoserine preference was observed (Fig 2C, D, F and G). A series of phosphopeptides designed to test this confirmed that PP2A-B56 shows reduced dephosphorylation of phosphorylation sites with a proline in the +1 position (Fig 3C). However, the preference for basic amino acids N-terminal to the phosphorylation site could not be confirmed. Rather, *in vitro*, PP2A-B56 seems to dephosphorylate phosphorylation sites with basic, acidic, or non-charged amino acids in this position equally well and thus a proline in the +1 position seems to be the major determinant for PP2A-B56 activity *in vitro*. Finally, comparing a set of peptides distinguished only by the presence of either phosphorylated serine or threonine amino acid residues revealed no particular difference in preference by PP2A-B56 toward these phosphorylation sites confirming the *in vivo* iceLogo representations (Fig 3D).

In conclusion, we find that the B56 and B55 regulatory subunits directly affect the phosphorylation site preference of their respective PP2A holoenzyme catalytic subunits, resulting in unique patterns of kinase opposition.

### Positional cues and substrate binding strength guide site-specific dephosphorylation by PP2A-B56

Our phosphoproteomic results revealed that most PP2A-B56-regulated phosphorylation sites are present on proteins containing LxxIxE docking motifs or on their direct interactors. This prompted

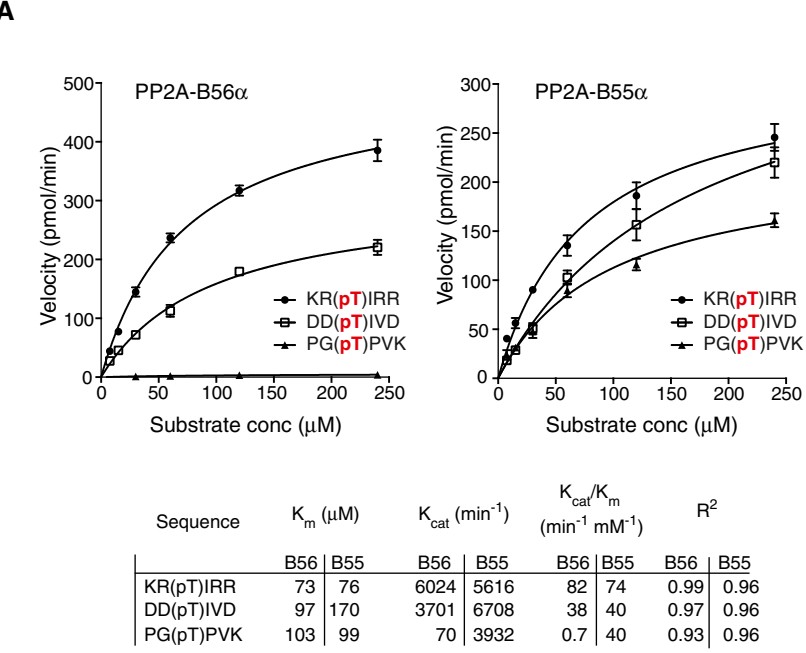

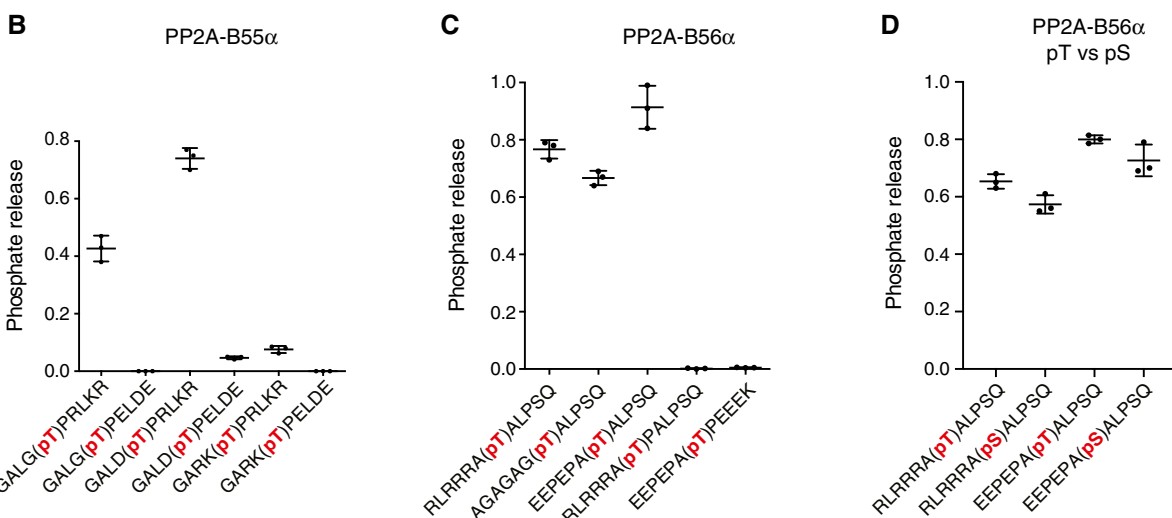

**Figure 3. Differential phosphorylation site preference of PP2A-B56 and PP2A-B55.**

A    Michaelis–Menten kinetic parameters of purified PP2A-B56α and PP2A-B55α holoenzymes were determined against the indicated phosphopeptides. Mean and standard deviation shown in plots as black bars ($n = 3$ independent experiments).

B–D    *In vitro* dephosphorylation by the PP2A-B55α and PP2A-B56α holoenzymes of panels of phosphorylated peptides as indicated. Mean and standard deviation shown in plots as black bars ($n = 3$ independent experiments).

deeper investigation into how the position or affinity of an LxxIxE motif in relation to a phosphorylation site might contribute to the observed differential dephosphorylation by PP2A-B56.

For this purpose, we turned to the transcription factor FoxO3, which binds to PP2A-B56 through a conserved LxxIxE motif (residues 448-MQTIPE-453; Fig 4A; Hertz *et al*, 2016). Consistently, FoxO3 S413 and S425 were among the up-regulated phosphorylation sites in the G1/S phosphoproteomic dataset (Table EV3).

Furthermore, dephosphorylation of FoxO3 by PP2A-B56 in an LxxIxE-dependent manner was previously observed to promote its nuclear translocation (Singh *et al*, 2010; Hertz *et al*, 2016).

First, a YFP-FoxO3 construct where the endogenous B56 LxxIxE docking site is mutated (FoxO3 2A) was generated. In this construct, motifs of low (L; $K_D \approx 17~\mu M$), intermediate (I; $K_D \approx 1.2~\mu M$), or high (H; $K_D \approx 0.08~\mu M$) affinity for PP2A-B56 were fused to the C-terminus of FoxO3 (Fig 4A). The ability of the engineered FoxO3

constructs to interact with PP2A-B56 was compared to that of wild-type (wt) FoxO3 and 2A. The L-C, I-C, and H-C FoxO3 variants bound approximately 0.5-, 4-, or 10-fold more PP2A-B56 than wt FoxO3, whereas no binding to PP2A-B56 was observed for FoxO3 2A (Fig 4B). Thus, binding between FoxO3 and PP2A-B56 was preserved when engrafting motifs onto ectopic FoxO3 sites allowing for the investigation of positional and compositional B56 motif-phosphorylation site relationships.

The engineered FoxO3 variants were expressed in HeLa cells and probed with a commercially available phosphoantibody against pS413 or a phosphoantibody produced in-house against pS253 (see Fig EV3A for validation of the pS253 antibody). The wt FoxO3 showed decreased levels of phosphorylation on S413 and S253 compared to FoxO3 2A as expected (Fig 4C and D). Dephosphorylation of pS413 and pS253 in L-C is comparable to that observed for FoxO3 2A. Furthermore, even though FoxO3 I-C binds approximately fourfold more PP2A-B56 than wt FoxO3, only a slightly reduced or similar dephosphorylation pattern of pS413 and pS253 is observed when compared to wt FoxO3 (Fig 4C and D). Even though distance in the primary amino acid sequence does not necessarily translate to distance in the tertiary structure, these results may reflect that at least S413 is closer to the endogenous FoxO3 LxxIxE motif (amino acids 448–453) compared to the motifs moved to the C-terminus (amino acid 673). On the other hand, the FoxO3 H-C variant, which binds roughly 10-fold more PP2A-B56 than wt FoxO3, showed a comparable dephosphorylation pattern with respect to pS413 and increased dephosphorylation of S253. These observations suggest that the decrease in dephosphorylation from moving PP2A-B56 motifs out of their natural context (L-C) can be partially (I-C) or fully (H-C) compensated for by motifs with increasing binding strength.

Phosphorylation of FoxO3 at T32, S253, and S644 promotes its retention in the cytoplasm (Brunet *et al*, 1999). Various extracellular stimuli lead to dephosphorylation by PP2A-B56 and subsequent translocation to the nucleus (Brunet *et al*, 1999; Hu *et al*, 2004; Singh *et al*, 2010).

Consistently, the nuclear/cytoplasmic distribution ratio of the respective YFP-FoxO3 constructs correlated with the amount of bound PP2A-B56, with FoxO3 H-C localizing primarily to the nuclear compartment (Fig 4E). This suggests that the appropriate combination of motif position and binding affinity of the FoxO3 LxxIxE motif is important for the proper regulation of its nuclear/cytoplasmic distribution.

Next, we investigated whether these results could be recapitulated *in vitro*. To this end, we focused on the PP2A-B56 substrate Cdc20 that we also identified in the phosphoproteomic analyses (Table EV3; Lee *et al*, 2017; Fujimitsu & Yamano, 2020). We have previously shown that a short fragment of Cdc20 fused to GST (GST-Cdc20 49-78) can be utilized for efficient *in vitro* dephosphorylation (Hein *et al*, 2017; Ueki *et al*, 2019). Cdc20 49–78 contains three TP phosphorylation sites (T55, T59, and T70) that can be phosphorylated by Cdk1. This Cdc20 fragment was engineered to contain a PP2A-B56 binding LxxIxE motif at varying distances (1× = 12 aa, 2× = 70 aa, and 4× = 130 aa, with respect to T70) from the phosphorylation sites (Fig 4F). In addition, we replaced the LxxIxE motif with AxxAxA in the 1× construct as a control. While the dephosphorylation rates of Cdc20 1× and 2× were similar, Cdc20 4× exhibited slower dephosphorylation kinetics. To probe the

importance of binding affinity, the 1× GST-CDC20 fragment was engineered to contain either a higher ($K_D = 1$ μM) or lower ($K_D = 17$ μM) affinity LxxIxE motif (Fig EV3B). The lower affinity construct showed slower dephosphorylation kinetics than the corresponding higher affinity construct. Taken together, these *in vitro* and *in vivo* experiments do not favor a model where a strict three-dimensional positioning (key-in-lock model) of the B56 docking motif is the sole determinant for the phosphatase to dephosphorylate certain phosphosites on the substrate. If this was the case, moving the docking motif to structurally remote positions such as the C-terminus of FoxO3 should abolish dephosphorylation and this is not what is observed. Rather, the overall conclusion from these experiments is that a combination of motif position and binding strength is important parameters, which determine phosphatase activity toward specific phosphosites on the substrate.

## PP2A-B56 is a regulator of ADAM17 phosphorylation status and shedding activity

The PP2A-B56 phosphoproteomic screens identified phosphorylated T735 (1.75-fold change; B56 inhibitor WT/3A, $P < 0.0001$) and S791 (1.21-fold change; B56 inhibitor WT/3A, $P < 0.002$) of the essential transmembrane metalloproteinase ADAM17 as sites of PP2A-B56 dephosphorylation. Interestingly, T735 and S791 are in close proximity to a highly conserved putative LxxIxE motif located in the cytoplasmic C-terminal tail of ADAM17 (759-**M**DTIQE-764) (Fig 5A). This suggests a mechanistic basis for the observed activity of PP2A-B56 toward phosphorylated ADAM17. To interrogate this further, we performed ITC experiments using purified B56α and measured binding to a synthetic ADAM17 peptide containing the putative LxxIxE motif. We measured a $K_D$ of 7 μM, while no binding was detected to a peptide containing a point mutation (I762A) in the interaction motif. A peptide where the MDTIQE sequence was engineered for optimal binding (**LDTI-QEE**) showed a $K_D$ of 0.9 μM (Figs 5B and EV4A). In all the following experiments, we use the murine ortholog of ADAM17, but will refer to the corresponding human amino acid numbering throughout. To validate the ITC measurements in cells, constructs encoding full-length Myc-tagged ADAM17 wt, ADAM17 I762A, or the ADAM17 variant with improved binding motif (LEE) were transfected into a HeLa cell line stably expressing YFP-B56α. YFP-B56α was immunopurified using a YFP affinity column and binding monitored by an anti-Myc antibody in a subsequent WB assay (Fig 5C). Binding was observed for wt ADAM17, but not for the I762A mutant showing that ADAM17 encodes a *bona fide* PP2A-B56 binding motif in its cytoplasmic C-terminal tail. As expected, the LEE variant showed increased binding to PP2A-B56 compared to wt ADAM17. To establish a firm connection between PP2A-B56 binding and dephosphorylation of ADAM17, the C-terminal cytoplasmic part (V724-C827) of ADAM17 wt, I762A, and LEE ADAM17 were fused to GST, purified, and *in vitro*-phosphorylated with protein kinase A (PKA) (Fig EV4B). GST alone was not phosphorylated showing that the phosphosignal detected is specific for ADAM17 (Fig EV4B). Subsequent exposure of PKA-phosphorylated GST-ADAM17 (V724-C827) fragments to recombinant PP2A-B56α revealed that dephosphorylation of the I762A construct was slower and dephosphorylation of the LEE construct was faster when compared to wt GST-ADAM17 (Fig 5D). Thus, PP2A-B56 can

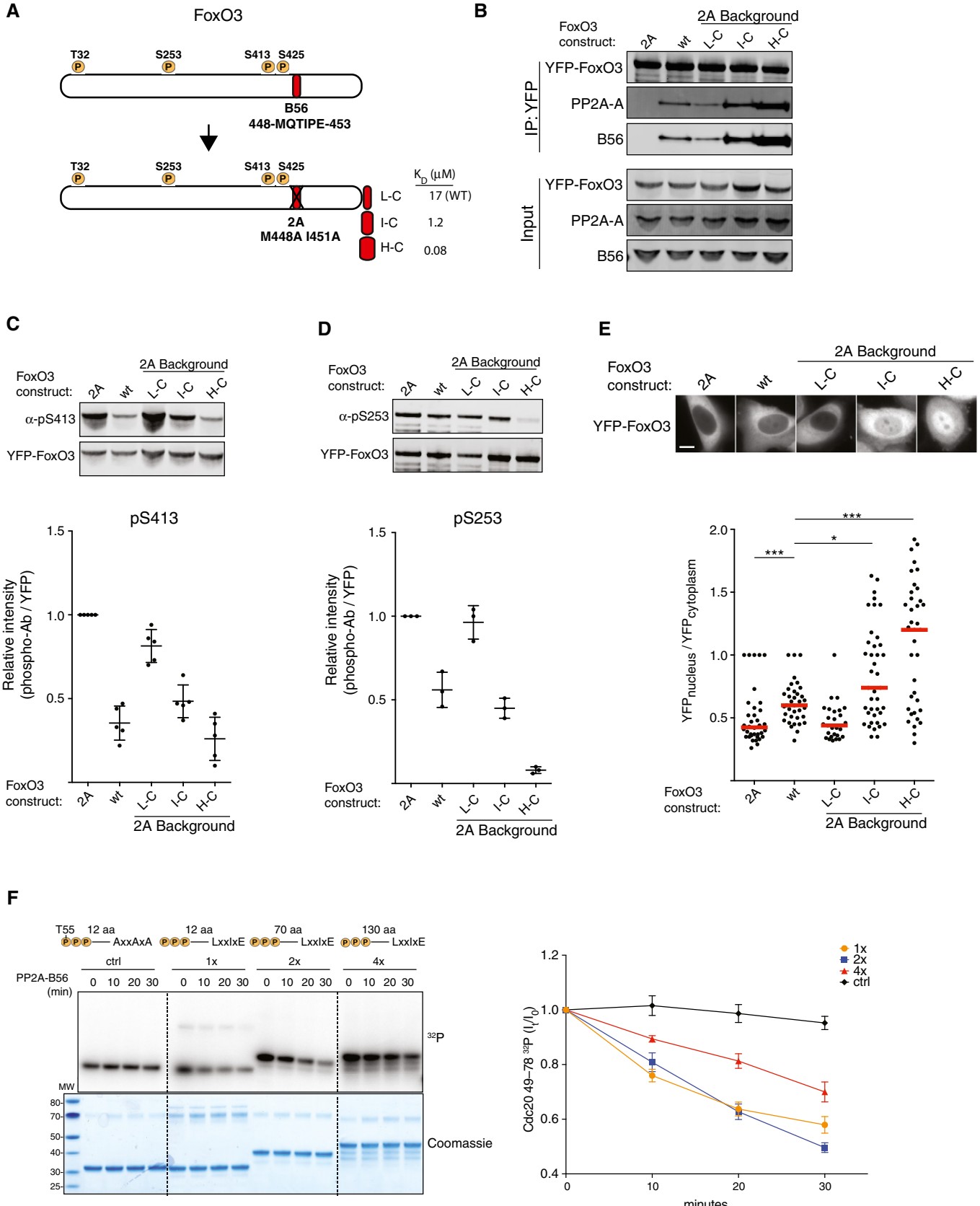

**Figure 4.**

**Figure 4. Mechanistic basis for site-specific dephosphorylation by PP2A-B56.**

A Schematics of engineered FoxO3 constructs. $K_D$ measurements of the LxxIxE motifs fused to the C-terminus of FoxO3 were performed previously (Hertz *et al*, 2016; Kruse *et al*, 2018).

B The indicated YFP-FoxO3 constructs were transfected into HeLa cells, purified using YFP resin and PP2A-B56 binding determined by Western blotting. PP2A-A; scaffold subunit.

C, D YFP-FoxO3 constructs were transfected into HeLa cells, and lysates (pS413) or YFP purifications (pS253) were subjected to Western blotting and probed with the indicated phosphoantibodies. Quantifications arise from five (pS413) or three (pS253) independent experiments. Mean and standard deviation shown in plots as black bars.

E Steady-state localization of the indicated YFP-FOXO3 variants in HeLa cells. Each data point represents quantification from a single cell. A representative result from at least three independent experiments is shown. At least 25 cells were counted per condition in the experiment shown. Scale bar is 10 μm. Mann–Whitney test significance values; *$P < 0.02$, ***$P < 0.0002$.

F Dephosphorylation by the PP2A-B56α holoenzyme complex of substrates with increasing length between phosphorylation sites and binding motifs as depicted. Engineered substrates containing three TP sites were phosphorylated with radioactive ATP using Cdk1 and incubated with the PP2A-B56α holoenzyme. Removal of radioactive phosphate was monitored over time. Mean and standard deviation from 3 experiments are shown.

regulate phosphorylation status of the ADAM17 C-terminal tail through binding to its LxxIxE motif.

In a process termed ectodomain shedding, the extracellular ADAM17 protease domain cleaves over 90 cell surface anchored proteins such as cytokines and growth factors leading to the release of their extracellular part as bioactive compounds (Zunke & Rose-John, 2017). ADAM17 shedding activity can be stimulated by phosphorylation of its intracellular C-terminal tail, and T735 and S808 phosphorylations enhance ADAM17-mediated shedding of epidermal growth factor receptor (EGFR) ligands (Soond *et al*, 2005; Xu & Derynck, 2010).

To probe the role of PP2A-B56 in regulating ADAM17 shedding activity, we established a genetic complementation system. Endogenous ADAM17 was removed using CRISPR/Cas9 technology in the colon cancer cell line DLD-1 allowing for functional complementation with exogenous ADAM17 (Fig EV4C–F). We established stable cell lines expressing ADAM17 wt, I762A, and LEE at equal levels that were correctly processed and equally transported to the cells surface (Figs 5E and EV4G). These cell lines were exposed to oxidative stress or X-ray irradiation, conditions known to induce ADAM17 activity, and shedding of the ADAM17 substrate amphiregulin (AREG) was monitored (Fig 5F). Interestingly, an inverse relationship between ADAM17 activity and PP2A-B56 binding was observed and this result was validated in an independent DLD-1 null clone (Fig EV4H).

Finally, we wanted to probe the effect of PP2A-B56 on the phosphorylation status of ADAM17 in this set-up. To this end, ADAM17 wt and I762A were immunopurified from cells exposed to oxidative stress. Mass spectrometry analysis of the ADAM17 band excised from the gel showed that in ADAM17 I762A, which does not bind PP2A-B56, phosphorylation of both T735 and S808 was up-regulated compared to ADAM17 wt (Fig 5G and Table EV7). Thus, PP2A-B56 regulates physiologically relevant phosphorylation sites on ADAM17 to modulate its shedding activity. We anticipate that PP2A-B56 when bound to ADAM17 regulates several phosphorylation sites in the C-terminal tail as well as the phosphorylation status of binding partners such as iRhom1/2 and that this collectively controls shedding.

### PP2A-B56 is a regulator of ADAM17-mediated growth factor signaling and tumorigenesis

Enhanced ADAM17 activity results in increased shedding of several epidermal growth factor receptor (EGFR) ligands and hyperactivation of EGFR signaling. Therefore, we tested the activation of EGFR signaling in the ADAM17 wt, I762A, and LEE cell lines as measured by EGFR Tyr1068 autophosphorylation after exposure to oxidative stress (Fig 6A). Strong binding between PP2A-B56 and ADAM17 correlated with low levels of EGFR autophosphorylation compared to the wt situation and the reverse was observed for the ADAM17 I762A cell line. Consistently, proliferation rates and invasion potential were decreased in the ADAM17 LEE cell line compared to the ADAM17 wt and ADAM17 I762A cell lines (Fig 6B and C).

Next, we investigated whether the binding of PP2A-B56 to ADAM17 influences *in vivo* tumor growth. As ADAM17-induced autocrine and paracrine signaling may influence the tumor microenvironment, we chose a syngeneic, orthotopic tumor model (DuPré *et al*, 2007). Similar to the DLD-1 cell system, we removed endogenous ADAM17 using CRISPR/Cas9 from the mouse breast cancer cell line 4T1 and exogenously expressed the different ADAM17 variants (Figs 6D, and EV4I and J). We then injected these cells into the mammary fat pad of BALB/c mice and monitored tumor growth. We found that tumors bearing the ADAM17 LEE variant grew significantly slower than ADAM17 I762A tumors (Fig 6E). Interestingly, none of the mice injected with ADAM17 LEE cells reached tumor endpoint criteria, as opposed to ADAM17 wt or I762A injected mice, which exhibited only 50% survival by the end of the experiment (Fig 6F).

Excessive ADAM17-mediated EGFR activation plays an important role in epithelial cancers (Ardito *et al*, 2012; Schmidt *et al*, 2018). Consistently, we show that PP2A-B56 binding to ADAM17 reduces EGFR Tyr1068 autophosphorylation and suppresses *in vivo* tumor growth.

## Discussion

From the perspective of basic PP2A biology, the most striking observation from our study is that the identity of the regulatory B-subunit affects the phosphorylation site preference of the holoenzyme catalytic subunit. Although previous work has hinted at this (Agostinis *et al*, 1992; Saraf *et al*, 2010), our global *in vivo* substrate mapping strategy and side-by-side comparison of two PP2A holoenzymes proves this to be a key basic principle of PP2A phosphatase function. This principle explains the need for an inhibitory mechanism toward PP2A-B55 during mitosis, while PP2A-B56 can remain active, as it has limited activity toward Cdk1 substrates, the major

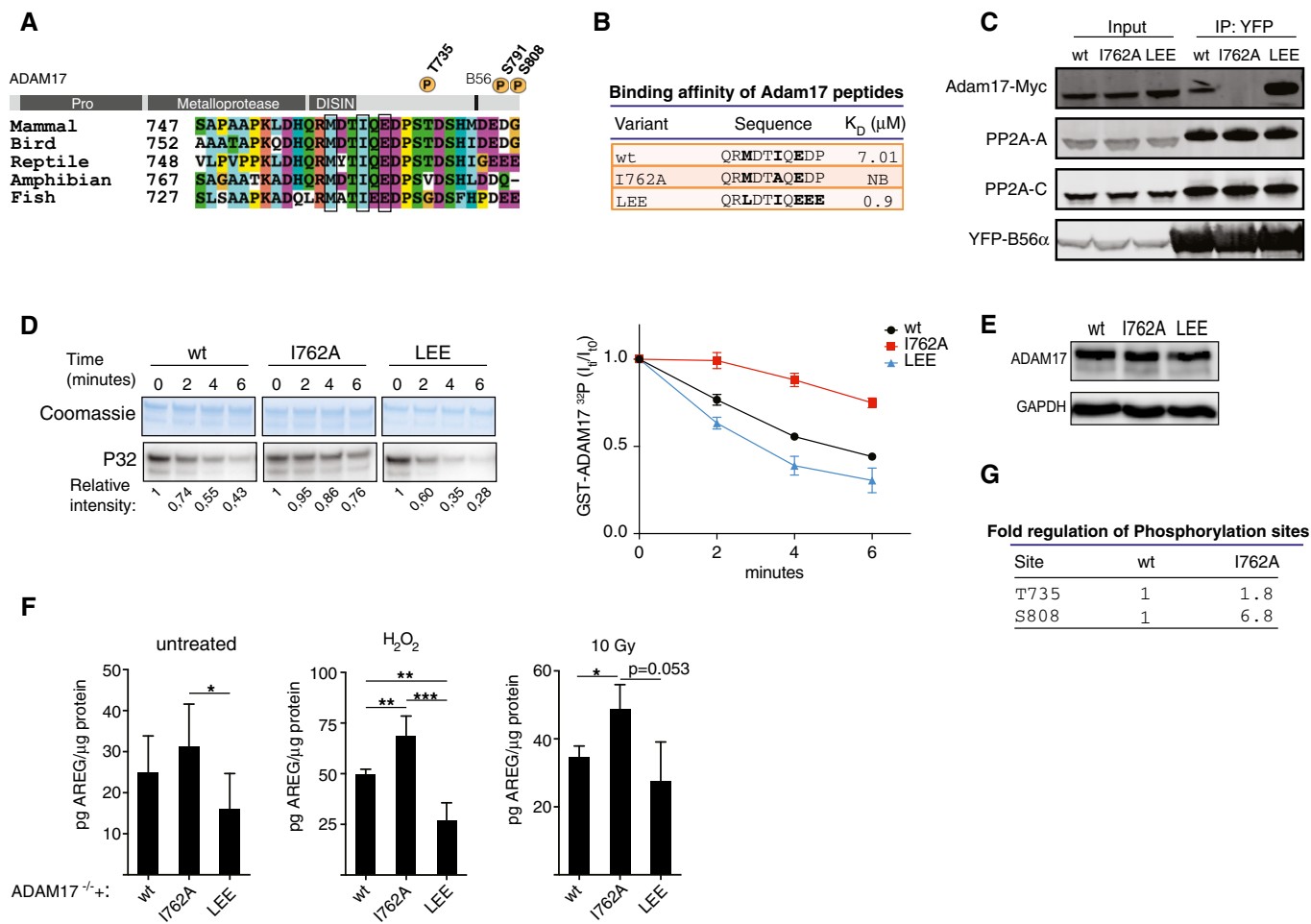

**Figure 5. PP2A-B56 is a regulator of ADAM17 phosphorylation status and shedding activity.**

A   Domain organization of ADAM17 and conservation of the PP2A-B56 binding motif.
B   $K_D$ values obtained by ITC measurements with full-length recombinant B56α and indicated ADAM17 variant peptides.
C   The indicated full-length murine Myc-ADAM17 derivatives were transfected into HeLa cells stably expressing YFP-B56α. YFP-B56α was purified (IP) and ADAM17 binding determined by Western blotting. PP2A-C; catalytic subunit, PP2A-A; scaffold subunit.
D   Dephosphorylation by the PP2A-B56α holoenzyme complex of the indicated phosphorylated GST-ADAM17 (V724-C827) fragments. The GST-ADAM17 (V724-C827) substrates were phosphorylated with radioactive ATP using protein kinase A and incubated with the PP2A-B56α holoenzyme. Removal of radioactive phosphate was monitored over time. The mean and standard deviation of 4 independent experiments are shown.
E   Protein expression of ADAM17 variants in the DLD-1 Adam17$^{-/-}$ cell line, determined by Western blot. GAPDH was used as an internal loading control.
F   Amphiregulin (AREG) shedding measured by ELISA of conditioned media from untreated, $H_2O_2$ treated or irradiated with X-ray DLD-1 Adam17$^{-/-}$ cells (clone #1) expressing full-length ADAM17 variants (wt, I762A or LEE). Two-sided, unpaired Student's $t$-test was applied to test for significant differences *$P < 0.05$, **$P < 0.01$, ***$P < 0.001$. Mean and standard deviation indicated from at least three independent experiments.
G   Exogenous wt and I762A full-length ADAM17 was immunopurified from Adam17$^{-/-}$ cells treated with $H_2O_2$ and subjected to label-free LC-MS/MS to determine differential phosphorylation status of T735 and S808 (S811 in murine ADAM17).

proline-directed mitotic kinase. How the B-subunits can affect active site specificity will be an important question to solve and likely involves unique direct interactions with the catalytic subunit.

Based on the data presented here, we suggest a model for differential dephosphorylation of individual phosphorylation sites by PP2A-B56. First, the phosphatase is spatially positioned on LxxIxE containing protein complexes through binding to the motif. This establishes a gradient of phosphatase activity around accessible phosphorylation sites. Second, the active range of this gradient is determined by the binding strength of the motif. Finally, an additional regulatory layer to PP2A-B56 site-specific activity is achieved

by the preference for basophilic/acidophilic over proline-directed phosphorylation sites. This model likely applies to other members of the PPP phosphatase family that use SLiMs for binding to substrates. Strikingly, this resembles in many ways how kinases achieve specificity suggesting a common set of core selectivity principles for these enzymes.

We show how these principles and their integration with our phosphoproteomic data can be applied to identify novel direct PP2A-B56 targets as exemplified by ADAM17. Activation of ADAM17 involves multiple layers of regulation, of which phosphorylation of its C-terminus constitutes an important mechanism. Yet,

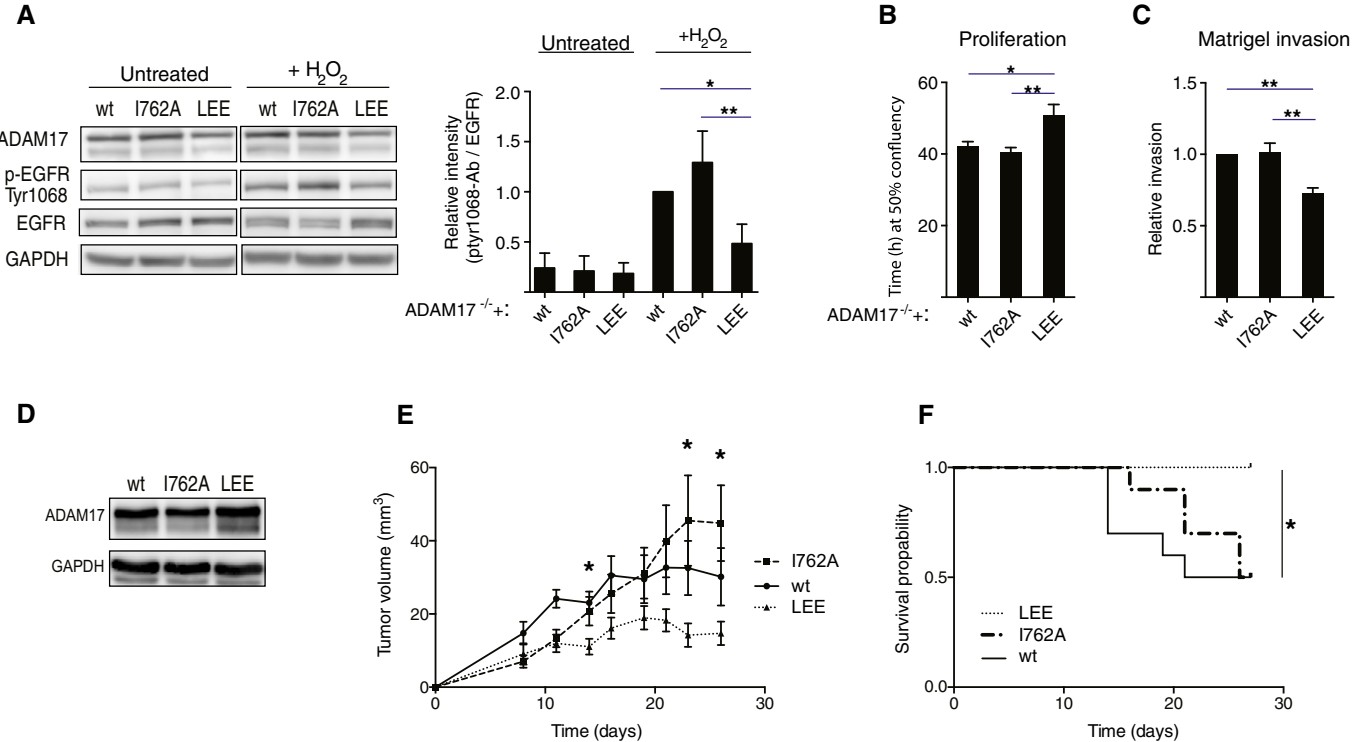

**Figure 6. PP2A-B56 is a regulator of ADAM17-mediated growth factor signaling and tumorigenesis.**

A   EGFR activation in the ADAM17 variant expressing DLD-1 Adam17$^{-/-}$ cells upon H$_2$O$_2$ treatment, determined by Western blot and quantified as the ratio of EGFR autophosphorylated at Tyr1068 to total EGFR. Two-sided, unpaired Student's *t*-test was applied to test for significant differences *P < 0.05, **P < 0.01. Mean and standard deviation indicated from three independent experiments.

B, C   (B) Cell proliferation and (C) Matrigel invasion of the ADAM17 variant expressing DLD-1 Adam17$^{-/-}$ cells. Two-sided, unpaired Student's *t*-test was applied to test for significant differences *P < 0.05, **P < 0.01. Mean and standard deviation indicated from three independent experiments.

D   Protein expression of the ADAM17 variants in the mouse breast cancer cell line 4T1 Adam17$^{-/-}$, determined by Western blot. GAPDH was used as an internal loading control.

E   Average tumor volume ± SEM of 4T1 Adam17$^{-/-}$ cells expressing the ADAM17 variants injected into the mammary fat pad of Balb/c mice (*n* = 10 for each group). Indicated significances are between the I762A and LEE tumors. Two-sided, unpaired Student's *t*-test was applied to test for significant differences *P < 0.05. Mean and standard error of the mean indicated.

F   Survival curve of the tumor bearing mice injected with 4T1 Adam17$^{-/-}$ cells expressing the ADAM17 variants (*n* = 10 for each group). The indicated significances are between the LEE and wt tumor bearing mice, and the LEE and I762A tumor bearing mice. Log-rank test was applied to test for significant differences. *P < 0.05.

how ADAM17 becomes deactivated is not clear. Here, we have revealed a novel inhibitory mechanism, whereby the PP2A-B56 holoenzyme reverts ADAM17 phosphorylations. We identified three PP2A-B56-regulated sites on ADAM17 (Thr735, Ser791, and Ser808), of which Thr735 and Ser808 have been shown to be phosphorylated in response to cellular stress and enhance ADAM17-mediated shedding of EGFR ligands (Xu & Derynck, 2010; Prakasam *et al*, 2014). In line with these findings, our cell-based assays confirmed that PP2A-B56 is a major regulator of stress-induced ADAM17 activity. Moreover, modulating the interaction between ADAM17 and PP2A-B56 had profound effects on *in vivo* tumor growth. Thus, it is tempting to speculate that at least one of the roles of PP2A-B56 as a tumor suppressor can be explained by its ability to restrict ADAM17-mediated EGFR signaling.

Collectively, our work provides an important foundation for understanding PP2A-regulated signaling in cells and future efforts aimed at interrogating specific pathways regulated by distinct PP2A holoenzymes.

# Materials and Methods

### Cell culture and reagents

Cancer cell lines DLD-1, HeLa, and 4T1 (provided by the Barbara Ann Karmanos Cancer Institute) cells were maintained in DMEM GlutaMAX containing 100 U/ml penicillin, 100 mg/ml streptomycin, and 10% FCS (all from Thermo Fisher Scientific). Stable HeLa cell lines were generated using the T-Rex doxycycline-inducible Flp-In system (Invitrogen) and cultivated like HeLa cells with the addition of 5 μg/ml blasticidin and 100 μg/ml hygromycin B. The DLD-1 and 4T1 cell lines were regularly tested for mycoplasma infection and authenticated by STR profiling (Eurofins). *Escherichia coli* DH5α and BL21(DE3) strains were maintained and propagated using standard microbiological procedures. The following drug concentrations were used: thymidine 2.5 mM, nocodazole 150 ng/μl, and doxycycline 10 ng/ml unless otherwise stated.

## CRISPR/Cas9 gene editing

DLD-1 Adam17 knockout cells were generated using the CRISPR-Cas9 system. Guide RNAs (gRNA) were designed using the WTSI genome editing tool (Hodgkins *et al*, 2015) and individually inserted in the vector pSpCas9(BB)-2A-GFP, as described (Ran *et al*, 2013). To determine the gRNA editing efficiency, cells were transfected with the pSpCas9(sgRNA)-2A-GFP vectors and verified by Indel Detection by Amplicon Analysis (IDAA) (Lonowski *et al*, 2017). Human gRNA 5 AAAGCGAGTACACTGTAAAA-3, targeting exon 4, and murine gRNA 5-ACAAAACTTGAGAGTCGTGG-3, targeting exon 3, showed the highest editing efficiencies and were subsequently transfected into DLD-1 and 4T1 cells, respectively. GFP-positive cells were single cell-sorted, expanded, and tested by qPCR and Western blot for Adam17 knockout. Additionally, we screened positive clones for bi-allelic frameshifts using Sanger sequencing (Eurofins).

## Expression constructs and cell line generation

Standard cloning techniques were used throughout. Point mutations were introduced by whole plasmid PCR. All constructs were fully sequenced. Synthetic DNA encoding the various B56 inhibitor sequences was purchased from GeneArt, Life Technologies. The pcDNA3.1 plasmid containing the murine cDNA of ADAM17 was provided by Prof. Dr. Stefan Rose-John (Kiel University, Kiel, Germany), and the I765A and M762L, D768E, P769E (LEE) mutants were generated by site-directed mutagenesis. ADAM17 wt, I765A, and LEE inserts were cloned into the sleeping beauty transposon vector v359 (Kowarz *et al*, 2015) and transfected into the DLD-1 A17$^{-/-}$ or 4T1 A17$^{-/-}$ cells using Fugene HD (Promega), and selected with 2 µg/ml puromycin (Sigma) for 6 days. For induction of ADAM17 expression, 10 ng/ml doxycycline (Sigma) was added to the cell culture medium and incubated for 24 h. Detailed mutagenesis and cloning strategies are available upon request.

## Antibodies

The following antibodies were used at the indicated dilutions: c-Myc (9E10, sc-40, 1:1,000, Santa Cruz Biotechnology), Separase (A302-215A, 1:2,000, Bethyl Laboratories), KIF4A (A301-074A, 1:1,000, Bethyl Laboratories), Axin1 (#2087, 1:1,000, Cell Signaling Technology), B56α (610615, 1:3,000, BD Biosciences), BubR1 (A300-995A,1:1,000, Bethyl Laboratories), PP2A catalytic subunit (05-421, 1:2,000, Millipore), PP2A scaffold subunit (#2041, 1:1,000, Cell Signaling Technology) GFP (ab290, 1:4,000, Abcam), FoxO3 pS413 (#8174, 1:1,000, Cell Signaling Technology), FoxO3 rabbit polyclonal α-pS253 (Raised against peptide CAPRRRAV(pS)MDNS; 1:500, Moravian Biotechnology), Anti-RFP (1:1,000; MBL, FM005), Anti-GFP (1:1,000; Roche, 11814460001), Rabbit anti-ADAM17 (1:1,000, Abcam, 2051), Rabbit anti-ADAM17 (1:1,000, Abcam, 39162), Rabbit anti-EGFR (1:1,000, cell signaling, 2232), Rabbit anti-pEGFR Y1068 (1:1,000, cell signaling, 2234), Mouse anti-Transferrin receptor (1:1,000, Invitrogen, 136800), Mouse anti-GAPDH (1:5,000, Sigma, G8795), rabbit anti-GFP (1:3,000, Takara Bio Clontech, 632592), Donkey anti-rabbit-HRP (1:2,000, GE Healthcare, NA934), Sheep anti-mouse-HRP (1:2,000, GE Healthcare, NXA931), and H3pS10 (06-570, 1:1,000, Millipore).

## Protein expression and purification

Cdc20 and ADAM17 fusion proteins were cloned into pGEX-4T-1 to generate N-terminally GST-tagged fusion proteins. Constructs were transformed into BL21 (DE3) cells, and expression was induced by addition of 0.5 mM IPTG at 18 degrees overnight. Following resuspension in buffer L (50 mM Tris pH 7.5; 300 mM NaCl; 10% glycerol; 0.5 mM TCEP, 1× complete EDTA-free tablets (Roche)), the sample was lysed using a high pressure homogenizer (Avestin). Lysate was clarified by centrifugation and loaded on a GSTrap HP 5 ml column and washed with buffer L to baseline absorbance at 280 nm. The proteins were eluted with buffer L containing 20 mM glutathione using a 20 CV gradient and the peak fractions pooled and collected. The peak fractions were concentrated on a vivaspin 20 and loaded on a Superdex 75 16/60 GL column equilibrated with buffer GF (50 mM NaP pH = 7.5; 150 mM NaCl; 10% glycerol; 0.5 mM TCEP). Relevant fractions were pooled and flash-frozen and stored at −80.

The B56 inhibitors and ctrl inhibitor were cloned into the pET30 expression vector and expressed in *E. coli* BL21 (DE3) cells as described above. Cells were suspended in buffer U (100 mM Tris pH = 8.0; 300 mM NaCl; 7.4 M urea) and sonicated followed by centrifugation for 2 × 25 min and sample filtered. The urea concentration was lowered to 1 M by sequential dialysis (6 M-3 M-1 M urea) followed by centrifugation. The lysate was filtered and applied to a 5 ml Ni-NTA affinity column and bound proteins eluted with an imidazole gradient. Peak fractions as measured by UV spectroscopy were pooled and the NaCl concentration lowered to 50 mM NaCl and applied to a MonoQ column. A gradient from 50 to 1,000 mM NaCl was applied and peak fractions collected and pooled. Subsequently, the pool was run on a Superdex 75 16/60 column equilibrated with (50 mM Tris pH = 7.5, 100 mM NaCl) and peak fractions pooled. Recombinant B56α, GST-Arpp19, and MASTL were produced as described previously (Hertz *et al*, 2016; Hein *et al*, 2017).

## Isothermal titration calorimetry

Recombinant B56α and B56 inhibitors were extensively dialyzed against ITC buffer 50 mM sodium phosphate pH 7.5, 200 mM NaCl, 0.5 mM TCEP. All experiments were performed on an Auto-iTC200 (Malvern Panalytical) instrument at 25°C. The B56α inhibitors were loaded into the syringe and titrated into the calorimetric cell containing B56α (direct titrations) or vice versa (reverse titrations). Control experiments with either the inhibitors or B56α injected in the sample cell filled with buffer were carried out under the same experimental conditions. These control experiments showed negligible heats of dilution in all cases. The titration sequence consisted of a single 0.4 µl injection followed by 19 injections, 2 µl each, with 150 s spacing between injections to ensure that the thermal power returns to the baseline before the next injection. The stirring speed was 750 rpm. Couple of ITC binding isotherms for direct and reverse titrations were globally fitted to a 1:1 model using the AFFINImeter software thus yielding a single set of binding parameters per interaction.

## Live-cell imaging

Live-cell analysis was performed on a DeltaVision Elite system using a ×40 oil objective with a numerical aperture of 1.35 (GE Healthcare). The DeltaVision Elite microscope was equipped with a CoolSNAP HQ2 camera (Photometrics). Cells were seeded in eight-well Ibidi dishes (Ibidi) and before filming, the media was changed to Leibovitz's L-15 (Life Technologies). Appropriate channels were recorded for the times indicated. For transient transfections, DNA constructs were transfected into HeLa cells using Lipofectamine 2000 (Life Technologies) 24 h prior to analysis. The nuclear/cytoplasmic distribution of YFP-FoxO3 was analyzed using SoftWoRx (GE Healthcare) software.

## Immunoprecipitations and displacement assay

Hela-FRT cell lines stably expressing mCherry wt or control (3A) B56 inhibitor were transfected with YFP-B56α. Cells were arrested in thymidine (2.5 mM) for 24 h and released into nocodazole (200 ng/ml) for 18 h. The expression of the inhibitors was induced 24 h prior of collection with the addition of 4 ng/ml doxycycline. Cells were collected by mitotic shake-off and lysed in low salt lysis buffer (50 mM NaCl, 50 mM Tris pH 7.4, 1 mM EDTA, 1 mM DTT, 0.1% NP40) supplemented with protease and phosphatase inhibitors (Roche) for 25 min on ice. Lysates were cleared for 15 min at 20,000 rcf and incubated with 20 μl pre-equilibrated GFP-trap beads (ChromoTek) for 45 min at 4°C. Following three washes with lysis buffer, the beads were eluted in 25 μl 2× loading buffer, boiled for 5 min, and separated by SDS–PAGE or subjected to quantitative Mass spectrometry as described in the label-free LC-MS/MS analysis section.

## Cell cycle synchronization for G1/S and M phosphoproteomics analyses

Stable HeLa Flp-In cells expressing either wild-type or mutant B56 inhibitor were arrested in either G1/S or mitotic phase of the cell cycle. For G1/S arrest, cells were treated with 2 mM thymidine for 24 h with the induction of wild-type or mutant B56 inhibitor for the last 12 h using 10 ng/ml doxycycline. A mitotic arrest was achieved by treating the cells with 2 mM thymidine for 24 h, followed by release into media containing 100 ng/ml nocodazole and 10 ng/μl doxycycline and collected by mitotic shake-off. Cells were washed once with PBS and snap-frozen. The amino acid sequence of the 4 × LxxIxE inhibitor is as follows: TGSTGSTGSTGSTGSLPRSST**LP TIHE**EEELSLCTGSTGSTGSTGSTGSLPRSST**LPTIHE**EEELSLCTGSTGST GSTGSTGSLPRSST**LPTIHE**EEELSLCTGSTGSTGSTGSTGSLPRSST**LP TIHE**EEELSLC. The corresponding 4 × control inhibitor sequence is as follows: TGSTGSTGSTGSTGSLPRSST**APTAHA**EEELSLCTGSTGST GSTGSTGSLPRSST**APTAHA**EEELSLCTGSTGSTGSTGSTGSLPRSST**AP TAHA**EEELSLCTGSTGSTGSTGSTGSLPRSST**APTAHA**EEELSLC.

## Label-free LC-MS/MS analysis

Pull-downs were analyzed on a Q-Exactive Plus quadrupole Orbitrap mass spectrometer (Thermo Scientific) equipped with an Easy-nLC 1000 (Thermo Scientific) and nanospray source (Thermo Scientific). Peptides were resuspended in 5% methanol/1% formic

acid and loaded on to a trap column [1 cm length, 100 μm inner diameter, ReproSil, $C_{18}$ AQ 5 μm 120 Å pore (Dr. Maisch, Ammerbuch, Germany)] vented to waste via a micro-tee and eluted across a fritless analytical resolving column (35 cm length, 100 μm inner diameter, ReproSil, $C_{18}$ AQ 3 μm 120 Å pore) pulled in-house (Sutter P-2000, Sutter Instruments, San Francisco, CA) with a 45-min gradient of 5–30% LC-MS buffer B (LC-MS buffer A: 0.0625% formic acid, 3% ACN; LC-MS buffer B: 0.0625% formic acid, 95% ACN). The Q-Exactive Plus was set to perform an Orbitrap MS1 scan ($R = 70K$; AGC target = 1e6) from 350 to 1,500 m/z, followed by HCD MS2 spectra on the 10 most abundant precursor ions detected by Orbitrap scanning ($R = 17.5K$; AGC target = 1e5; max ion time = 50 ms) before repeating the cycle. Precursor ions were isolated for HCD by quadrupole isolation at width = 1 m/z and HCD fragmentation at 26 normalized collision energy (NCE). Charge state 2, 3, and 4 ions were selected for MS2. Precursor ions were added to a dynamic exclusion list ± 20 ppm for 15 s. Raw data were searched using COMET (release version 2014.01) in high-resolution mode (Eng *et al*, 2013) against a target-decoy (reversed) (Elias & Gygi, 2007) version of the human proteome sequence database (UniProt; downloaded 2/2013, 40,482 entries of forward and reverse protein sequences) with a precursor mass tolerance of ± 1 Da and a fragment ion mass tolerance of 0.02 Da, and requiring fully tryptic peptides (K, R; not preceding P) with up to three mis-cleavages. Static modifications included carbamidomethylcysteine, and variable modifications included oxidized methionine. Searches were filtered using orthogonal measures including mass measurement accuracy (± 3 ppm), Xcorr for charges from +2 through +4, and dCn targeting a < 1% FDR at the peptide level. Quantification of LC-MS/MS spectra was performed using MassChroQ (Valot *et al*, 2011) and the iBAQ method (Schwanhäusser *et al*, 2011). Missing values were imputed from a normal distribution in Perseus to enable statistical analysis and visualization by volcano plot (Tyanova *et al*, 2016). Statistical analysis was carried out in Perseus by two-tailed Student's *t*-test.

## Phosphoproteomics analysis

Cell pellets were lysed in ice-cold lysis buffer [8 M urea, 25 mM Tris–HCl pH 8.6, 150 mM NaCl, phosphatase inhibitors (2.5 mM beta-glycerophosphate, 1 mM sodium fluoride, 1 mM sodium ortho-vanadate, 1 mM sodium molybdate) and protease inhibitors (1 mini-Complete EDTA-free tablet per 10 ml lysis buffer; Roche Life Sciences)] and sonicated three times for 15 s each with intermittent cooling on ice. Lysates were centrifuged at 15,000 × *g* for 30 min at 4°C. Supernatants were transferred to a new tube, and the protein concentration was determined using a BCA assay (Pierce/Thermo Fisher Scientific). Equal protein amounts are carried forward for analysis. For reduction, DTT was added to the lysates to a final concentration of 5 mM and incubated for 30 min at 55°C. Afterward, lysates were cooled to room temperate and alkylated with 15 mM iodoacetamide at room temperature for 45 min. The alkylation was then quenched by the addition of an additional 5 mM DTT. After sixfold dilution with 25 mM Tris–HCl pH 8, the samples were digested overnight at 37°C with 1:100 (w/w) trypsin. The next day, the digest was stopped by the addition of 0.25% TFA (final v/v), centrifuged at 3,500 × *g* for 15 min at room temperature to pellet

precipitated lipids, and peptides were desalted. Peptides were lyophilized and stored at −80°C until further use.

Phosphopeptide purification was performed as previously described (Kettenbach & Gerber, 2011). Briefly, peptides were resuspended in 2 M lactic acid in 50% ACN ("binding solution"). Titanium dioxide microspheres were added and vortexed by affixing to the top of a vortex mixer on the highest speed setting at room temperature for 1 h. Afterward, microspheres were washed twice with binding solution and three times with 50% ACN/0.1% TFA. Peptides were eluted twice with 50 mM $KH_2PO_4$ (adjusted to pH 10 with ammonium hydroxide). Peptide elutions were combined, quenched with 50% ACN/5% formic acid, dried, and desalted.

Phosphopeptides were resuspended in 133 mM HEPES (SIGMA) pH 8.5, and TMT reagent (Thermo Fisher Scientific) stored in dry acetonitrile (ACN) (Burdick & Jackson) was added, vortexed to mix reagent and peptides. After 1 h at room temperature, an aliquot from each channel was withdrawn to check for labeling efficiency, while the remaining reaction was stored at −80°C. Once labeling efficiency was confirmed to be at least 95%, each reaction was quenched with ammonium bicarbonate for 10 min, mixed, acidified with 20% TFA, and desalted. The desalted multiplex was dried by vacuum centrifugation and separated by offline pentafluorophenyl (PFP)-based reversed-phase HPLC fractionation as published (Grassetti et al, 2017). Briefly, TMT-labeled phosphopeptides were separated over a gradient of 5–55% Buffer B from 0 to 61 min. Forty-eight fractions were collected and concatenated into 24 by mixing the nth and $n^{th} + 24^{th}$ fraction. Buffer B: 95% ACN/0.1% TFA; Buffer A: 3% ACN/0.1% TFA.

TMT-labeled samples were analyzed on an Orbitrap Fusion (Senko et al, 2013) mass spectrometer (Thermo Scientific) equipped with an Easy-nLC 1000 (Thermo Scientific). Peptides were resuspended in 8% methanol/1% formic acid and loaded onto a column (45 cm length, 100 μm inner diameter, ReproSil, $C_{18}$ AQ 1.8 μm 120 Å pore) pulled in-house across a 2-h gradient from 3% acetonitrile/0.0625% formic acid to 37% acetonitrile/0.0625% formic acid. The Orbitrap Fusion was operated in data-dependent, SPS-MS3 quantification mode (Ting et al, 2011; McAlister et al, 2014) wherein an Orbitrap MS1 scan was taken (scan range = 350–1,200 m/z, $R$ = 120K, AGC target = 3e5, max ion injection time = 100 ms), followed by data-dependent Orbitrap MS2 scans of the most abundant precursors for 3 s: ion selection; charge state = 2: minimum intensity 2e5, precursor selection range 650–1,200 m/z; charge state 3: minimum intensity 3e5, precursor selection range 525–1,200 m/z; charge states 4 and 5: minimum intensity 5e5; quadrupole isolation = 0.7 m/z, $R$ = 30K, AGC target = 5e4, max ion injection time = 80 ms, CID collision energy = 32%; and Orbitrap MS3 scans for quantification ($R$ = 50K, AGC target = 5e4, max ion injection time = 100 ms, HCD collision energy = 65%, scan range = 110–750 m/z, synchronous precursors selected = 5). The raw data files were searched using COMET with a static mass of 229.162932 on peptide N-termini and lysines and 57.02146 Da on cysteines, and a variable mass of 15.99491 Da on methionines and 79.96633 Da on serines, threonines, and tyrosines against the target-decoy version of the human proteome sequence database (UniProt; downloaded 2/2013, 40,482 entries of forward and reverse protein sequences) and filtered to a < 1% FDR at the peptide level. Quantification of LC-MS/MS spectra was performed using in-house developed software. Phosphopeptide intensities were adjusted based

on total TMT reporter ion intensity in each channel to adjust for mixing errors and $\log_2$-transformed. *P*-values were calculated using a two-tailed Student's *t*-test assuming unequal variance.

## Inhibition of PP2A in lysates

Recombinant GST-Arpp19 wt or S62A mutant (235 μg) was incubated with ~ 6 μg purified MASTL/Greatwall and thio-ATP (Tocris) in kinase buffer (50 mM Tris–HCl pH 7.5, 10 mM $MgCl_2$, 0.1 mM EDTA, 2 mM DTT, 0.01% Brij 35) for 2 h at 30°C. The GST Arpp19 proteins were concentrated, size-exclusion chromatography was performed on a Superdex 200 10/300 column, and the buffer was exchanged to 300 mM NaCl, 50 mM Tris pH 8, 8.7% glycerol. Peak fractions were collected, pooled, and concentrated. Aliquots were snap-frozen in liquid nitrogen.

Following double thymidine synchronization, cells were released into nocodazole 200 ng/ml for 16 h. Mitotic cells were collected by shake-off and counted. Lysates from $5 \times 10^6$ cells were analyzed per condition. Cells were lysed in lysis buffer (150 mM NaCl, 50 mM Tris pH 7.4, 0.1% NP40, 1 mM DTT, supplemented with EDTA-free protease inhibitors (Roche)) plus 50 μg of the corresponding PP2A inhibitors. Full-length thiophosphorylated Arpp19 WT or S62A was used for the inhibition of PP2A-B55. A high-affinity LxxIxE peptide (WLPRSSTLPTIHEEEELSLC) or control peptide (WLPRSSTLPTA-HADSVLSLC) was used to inhibit PP2A-B56 specifically. Cells were lysed for 5 or 15 min at 30°C while shaking and the reactions were stopped by the addition of lysis buffer supplemented with 2xPhosStop tablets (Roche). Lysates were cleared for 15 min, 20,000 rcf at 4°C, and snap-frozen in liquid nitrogen.

## ARPP19 and B56 peptide lysate phosphoproteomics analysis

Lysates were precipitated by adding 4× volume of ice-cold acetone (Burdick & Jackson) and frozen for 1 h at −20°C. Afterward, samples were centrifuged at $14,800 \times g$ for 15 min at 4°C to pellet precipitated proteins. Pellets were washed twice with ice-cold acetone and dried. Proteins were resuspended in urea lysis buffer as described above. Phosphopeptide enrichment was carried out using the Fe-NTA phosphopeptide enrichment kit (Thermo Fisher) according to the manufacturer's instructions. Phosphopeptides were labeled with TMT reagents as described above and offline separated. TMT-labeled samples were analyzed on an Orbitrap Fusion Lumos mass spectrometer (Thermo Scientific) equipped with an Easy-nLC 1200 (Thermo Scientific). Peptides were resuspended in 8% methanol/1% formic acid and loaded onto a column (45 cm length, 100 μm inner diameter, ReproSil, $C_{18}$ AQ 1.8 μm 120 Å pore) pulled in-house across a 2-h gradient from 3% acetonitrile/0.0625% formic acid to 37% acetonitrile/0.0625% formic acid. The Orbitrap Lumos was operated in data-dependent, SPS-MS3 quantification mode (Ting et al, 2011; McAlister et al, 2014) wherein an Orbitrap MS1 scan was taken (scan range = 350–1,250 m/z, $R$ = 120K, AGC target = 2.5e5, max ion injection time = 50 ms), followed by data-dependent Orbitrap MS2 scans of the most abundant precursors for 2 s: ion selection; charge state = 2: minimum intensity 2e5, precursor selection range 650–1,250 m/z; charge state 3: minimum intensity 3e5, precursor selection range 525–1,250 m/z; charge states 4 and 5: minimum intensity 5e5; quadrupole isolation = 1 m/z, $R$ = 30K, AGC target = 5e4, max ion injection time = 55 ms, CID

collision energy = 35%; and Orbitrap MS3 scans for quantification ($R$ = 50K, AGC target = 5e4, max ion injection time = 100 ms, HCD collision energy = 65%, scan range = 100–500 m/z, synchronous precursors selected = 5). The raw data files were searched, and data were processed as described above.

### In vitro phosphatase motif assay

293T Freestyle cells were transiently transfected with pEXPR-B56α, pCBS-KS+−2AAA, and pCBS-KS+PP2ACA. Forty-eight hours after transfection, cells were lysed in lysis buffer (50 mM Tris pH 7.5, 150 mM NaCl, and 1 mM MnCl₂) and sonicated three times for 15 s each with intermittent cooling on ice. Lysates were centrifuged at 15,000 × $g$ for 30 min at 4°C. Strep-Tactin Sepharose was added to the lysates and incubated while rotating for 2 h at 4°C. Beads were collected by centrifugation, washed three times with lysis buffer, and eluted with 1× Buffer E (Strep-Tactin elution buffer with desthiobiotin). Libraries of naturally occurring phosphopeptides were generated from 293T Freestyle cells. Cells were lysed in urea lysis buffer as described above with the modification that after reduction and alkylation, the lysate was diluted 2.5-fold, and the protease Lys-C was added to a final concentration of 1:100 w/w. After overnight digest, peptides were desalted, and phosphopeptides were enriched and labeled with TMT reagents as described above. Individual pools of TMT-labeled phosphopeptides were dephosphorylated with PP2A-2AAA-B56α in the presence or absence of 10 nM calyculin A for 0.5, 1, 2, 3, or 4 h. Reactions were quenched with 0.1% TFA, mixed, desalted, dephosphorylated peptides were removed using TiO₂ as described above, and analyzed on an Orbitrap Fusion Lumos mass spectrometer as described above.

### In vitro phosphatase assays

*In vitro* phosphatase assays were performed with PP2A-B56α or PP2A-B55α, which were purified from HeLa cell extracts as described (Hein *et al*, 2017; Kruse *et al*, 2018). 15 µg of GST-fusion protein (GST-CDC20 49-78, engineered with different spacing or affinity of the LxxIxE motif) was incubated with CDK1-CyclinB1 (Sigma #SRP5009) in 50 µl reactions in kinase buffer (50 mM Tris–HCl pH 7.5, 10 mM MgCl₂, 0.1 mM EDTA, 2 mM DTT, 0.01% Brij 35) with 500 µM ATP and 1 µCi (γ-³²P)-ATP (PerkinElmer) at 30°C for 60 min. Reactions were stopped by the addition of 10 µM RO-3306 (Calbiochem). GST-ADAM17 (V724-C827) was phosphorylated with recombinant protein kinase A (New England Biolabs #P6000S). PD Spin Trap G25 columns (GE Healthcare) were used to exchange the buffer to phosphatase buffer (50 mM Tris pH 7.4, 1 mM MnCl₂, 1 mM DTT, 0.1% IGEPAL, 150 mM NaCl).

Non-stick tubes were pre-treated with blocking buffer (50 mM Tris pH 7.4, 0.1 mM MnCl₂, 1 mM MgCl₂, 1 mM DTT, 0.1% NP40, 300 mM NaCl, 2 mg/ml BSA) on ice for the dephosphorylation reactions. 70 ng of PP2A-B56 holoenzyme was added to 84 µl of phosphorylated substrate (~ 7 µg). Samples of ~ 1.75 µg were taken out at the indicated time-points, added to 4× SDS loading buffer and boiled for 5 minutes. Samples were separated by SDS–PAGE. Gels were dried, exposed for 3 days, and imaged on Typhoon FL 950 (GE Healthcare). Analyses and quantifications were carried out in ImageJ.

Michaelis–Menten kinetic parameters of the purified PP2A-B56α and PP2A-B55α holoenzymes were determined against the indicated phosphopeptides. Peptides were purchased from Peptide 2.0 Inc (Chantilly, VA, USA). The purity obtained in the synthesis was 98% as determined by high-performance liquid chromatography (HPLC) and subsequent analysis by mass spectrometry. Initial velocity ($V_0$) was determined at varying concentrations of substrate (7.5–240 µM) incubated with ~ 3 ng of the indicated PP2A holoenzyme in phosphatase buffer [50 mM Tris pH 7.4, 1 mM MnCl₂, 1 mM DTT, 0.1% (vol/vol) IGEPAL, 150 mM NaCl] for 4 min at 30°C. Release of inorganic phosphate was measured using the PiColorLock Phosphate Detection System (Expedon). Data from three independent experiments were fitted to the Michaelis–Menten model and kinetic parameters extracted using GraphPad Prism version 6.0e for Mac OS X.

### Data analysis

Phosphorylation site analysis was performed on phosphopeptides with the phosphorylation site localization score of 0.75 or higher. IceLogos were generated using singly phosphorylated sites with a phosphorylation localization score of 0.75 or higher (Colaert *et al*, 2009). For the network analysis, protein–protein interactions between previously identified B56 SLiM-containing proteins (Hertz *et al*, 2016; Wu *et al*, 2017) and proteins with significantly increased phosphorylation sites (log₂ ratio > 0.8 (1.75-fold), *P*-value < 0.05, phosphorylation site localization probability > 75%) were determined using the STRING database (Szklarczyk *et al*, 2017) and visualized in Cytoscape (Shannon *et al*, 2003).

### Amphiregulin shedding assay

Amphiregulin shedding experiments were performed according to the manufacturer's protocol (R&D systems). To evaluate the effect of the ADAM17-B56 binding, we induced ADAM17 expression in the DLD-1 A17wt, I762A, and LEE cell lines by changing to full medium containing 10 ng/ml doxycycline (Sigma) and collected the supernatant after 24 h (constitutive shedding). For induced shedding, the cells were treated with 600 µM H₂O₂ for 30 min (Merck), X-ray radiated with 10 Gy with a dose rate of 1 Gy/min using the X-ray generator CP160 (Faxitron X-Ray Corp.), or H₂O vehicle control for 30 min, and incubated for 3 h in full growth medium. Subsequently, the medium was changed and the supernatant collected after 1 h. The shedding results were normalized to the protein concentration to correct for possible differences in cell numbers.

### Proliferation assay

DLD-1 A17wt, I762A, and LEE cell lines were seeded at $1.5 \times 10^4$ cells per well in a 96-well plate in 200 µl full medium supplemented with 10 ng/ml doxycycline. The cells were subsequently incubated at 37°C and 5% CO₂ in an IncuCyte S3 (Sartorius) for 90 h, and confluency was recorded every 2 h. The timepoint 50% confluence was calculated from growing curves, which were created using the Incucyte ZOOM software (Sartorius).

### Invasion assay

Matrigel invasion assays were performed according to the manufacture's protocol. In short, ADAM17 expression was induced and

upon re-hydration of the pre-coated Matrigel invasion chambers (Corning), $2 \times 10^5$ cells were seeded in 500 µl FBS free DMEM (Gibco) in the upper chamber of the inserts. Next, the inserts were transferred into a 24-well plate containing 1 ml full DMEM and incubated for 24 h at 37°C. After incubation, invaded cells were fixed in 4% para-formaldehyde (Sigma), stained with 4% crystal violet (Sigma), and visualized under a light microscope (Axioplan 2, Zeiss). Invasion was determined by counting 10 randomly taken pictures (AxioCam, Zeiss) at a 10× magnification.

### Cell surface biotinylation assay

ADAM17 expression in the DLD-1 A17wt, I762A, and LEE cell lines was induced and 24 h later, cells were washed twice in cold PBS and incubated for 30 min with 0.5 mg/ml non-cleavable EZ-Link Sulfo-NHS-LC-Biotin (Thermo Scientific) in PBS. The biotinylation was quenched by washing three times with 100 mM glycine (Appli-Chem) in PBS and additionally three PBS washes. Cells were lysed for 30 min in RIPA buffer supplemented with protease inhibitors. Cell lysates were cleared by centrifugation, protein concentration equalized by BCA assay, and supernatants incubated with strepta-vidin-agarose beads (Sigma-Aldrich) for 2 h at 4°C. Beads were washed three times in RIPA and bound proteins released by heating 5 min at 95°C in 2× SDS–PAGE sample buffer. Samples were analyzed by Western blot.

### *In vivo* mouse model

Mice were randomly allocated into cages and mice within the same cage received the same treatment. On the day of injection, 4T1 A17wt, I762A, and LEE cells were harvested and $1 \times 10^4$ cells in 50 µl PBS containing 10 ng/ml doxycycline injected into the fourth mammary fat pat of 7-week-old female BALB/c mice (Janvier Labs). Sample size calculations performed with an alpha level of 0.05, 80–90% power, and an estimated difference between wt and LEE groups from previous experiments gave 8–11 mice/group. All mice were housed in ventilated cages in groups of 5 and maintained in a climate-controlled room at a temperature of $22 \pm 2°C$ and a relative humidity of $50 \pm 5\%$ under a 12-h light/dark cycle and fed a stan-dard diet and water *ad libitum*. Measurements of the primary tumor size using calipers and the mouse weight were monitored in a blinded fashion 2–3 times a week. The drinking water was supple-mented with 1 mg/ml doxycycline (Sigma) and 5% sucrose (Sigma) and changed every 2–3 days. All experiments were performed in accordance with authorization and guidance from the Danish Inspectorate for Animal Experimentation. The cell lines were tested negative for murine pathogens by IMPACT testing (IDEXX Laborato-ries).

### Quantification and statistical analysis

All statistical analyses were performed using GraphPad Prism version 6.0e for Mac OS X. Statistical details and definition of parameters can be found in figure legends. The statistical signifi-cance level was chosen as 0.05, but *P*-values for individual tests are indicated in the respective figure legends. Statistical methods were not employed to determine sample size or to determine whether the data met the assumptions of the statistical approach.

## Data availability

Mass spectrometry data have been deposited to ProteomeXchange PXD015205, http://proteomecentral.proteomexchange.org/cgi/Get Dataset?ID = PXD015205, MassIVE MSV000084245.

**Expanded View** for this article is available online.

### Acknowledgements

Work at the Novo Nordisk Foundation Center for Protein Research is supported by grant NNF14CC0001, and JN is supported by grants from the Danish Cancer Society (R167-A10951-17-S2), Independent Research Fund Denmark (DFF 8021-00101B and DFF 7016 00086), and Novo Nordisk Founda-tion (NNF18OC0053124). M.K was supported by the Danish Cancer Society: R146-A9211-16-S2. A.N.K was supported by grants from NIH/NIGMS (R35GM119455, P20GM113132). The Orbitrap Fusion Tribrid mass spectrometer was acquired with support from NIH (S10-OD016212). S.P.G. is funded by the European Union's Horizon 2020 research and innovation program under the Marie Sklodowska-Curie grant agreement no 798716. We thank the protein production platform at the Novo Nordisk Foundation Center for Protein Research for their help with producing recombinant proteins, the FACS facility at the Biotech Research and Innovation Centre (BRIC) for help on cell sorting, Morten Frödin, BRIC for advice on CRISPR/CAS9 gene knockout, and staff at the Department of Experimental Medicine animal facility for assistance on mouse experiments. Mass spectrometry data have been deposited to Proteo-meXchange PXD015205, MassIVE MSV000084245, password p730.

### Author contributions

TK did B56 inhibitor construction and characterization, FoxO3 work, and all *in vitro* dephosphorylation assays together with DHG. SPG did study design, performed, evaluated, or supervised functional assays, Western blots, and IP's and mouse experiments, and made the CRISPR/Cas9 knock-out and overexpression cells, analyzed the data. JS-P performed and contributed to the shedding assays and mouse experiments. HSP performed and contributed to proliferation and invasion assays. JS contrib-uted pilot data. DN generated the expression constructs. JBH contributed to B56 and B55 lysate phosphoproteomics. BL-M did all ITC experiments. EPTH contributed pilot data on the ADAM17-PP2A interaction. IN gener-ated the samples for, performed, and analyzed the in-cell phosphopro-teomics experiments. IN performed the B55 lysate phosphoproteomic analysis. DHG and ANK performed the B56 lysate phosphoproteomics experiment and analyzed the data. IN analyzed ADAM17 phosphorylation status. HN purified the protein. HN and ANK performed the *in vitro* phos-phatase motif assay. TK, MK, ANK, and JN conceived experiments, analyzed the data, and wrote the paper.

### Conflict of interest

The authors declare that they have no conflict of interest.

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
