## [Review Process File · The EMBO Journal]

Mechanisms of site-specific dephosphorylation and kinase opposition imposed by PP2A regulatory subunits

Thomas Kruse, Sebastian Gnosa, Isha Nasa, Dimitriya Garvanska, Jamin Hein, Hieu Nguyen, Jacob Samsøe-Petersen, Blanca Lopez-Mendez, Emil Hertz, Jeanette Schwarz, Hanna Pena, Denise Nikodemus, Marie Kveiborg, Arminja Kettenbach and Jakob Nilsson

Review timeline:

Submission date:	11 th October 2019
Editorial Decision:	13 th November 2019
Revision received:	5 th March 2020
Editorial Decision:	30 th March 2020
Revision received:	3 rd April 2020
Accepted:	21 st April 2020

Editor: Hartmut Vodermaier

Transaction Report:

1st Editorial Decision

13th November 2019

Thank you for submitting your manuscript on PP2A specificity principles to our editorial office. I have now received comments on it from three expert referees, copied below for your information. As you will see, the referees acknowledge the overall interest and quality of this work, but they also raise a number of concerns that would need to be addressed prior to acceptance. In particular, it will be critical to address the issue related to phosphosite localization confidence in the phosphoproteomics data, raised by the mass spectrometry expert referee 3. Furthermore, all referees consider the last section on ADAM17 still somewhat preliminary/superficial, making it clear that at least some further analysis on this part would be required, in addition to decreasing the emphasis on this aspect in the abstract. Similarly, the fact that the study focusses primarily on B56 and much less on B55 or other subunits also needs to be better reflected in title/abstract and throughout the paper. Finally, I agree with the referees that a more careful and explicit presentation will be important - including better experimental description, deposition/making available of datasets, careful proofreading/editing, and converting the currently rather dense four-figure format into the more extended/more accessible EMBO Journal article style.

Should you be able to satisfactorily address these key points as well as the other, more specific issues mentioned in all three reports, then we would be interested in considering a revised manuscript further for EMBO Journal publication.

REFeree REPORTS

Referee #1:

Kruse and al. investigate how PP2A B56 and PP2A B55 phosphatases target their substrates for

dephosphorylation. To do so, they have developed a strategy combining the use of specific phosphatase inhibitors and global mass spectrometry analysis to define which kind of phospho amino-acid sequences are recognised by the phosphatases. Their results suggest that regulatory subunits control substrate recognition at two different levels. The first interesting concept is that the regulatory subunits B56 or B55 modulate the PP2A catalytic subunit resulting in the recognition of different phosphomotifs. The second concept is based on the fact that B56 regulatory subunit recognises a docking site at the distance of the phosphosite, which contributes to substrate recognition. Using Cdc20 and Fox03 as model substrates, their work highlights the importance of the distance between the docking site and the phosphosite for efficient dephosphorylation. By moving the docking site at different places of the protein or by increasing the quantity of docking motifs on the substrate, they also show some flexibility suggesting that the binding strength (and not only the docking position) is important to determine which phosphosites are dephosphorylated. Finally, they show the importance of the B56 docking motif on the protein ADAM17 to negatively regulate ADAM17 activity and tumour growth in mice.

Several recent publications from this lab have significantly contributed to our knowledge about phosphatase specificity. Overall, the article is well written and easy to follow. The authors present the article as a side-by-side comparison of the two PP2A holoenzymes but in reality they put much more emphasis on B56 than B55, for which some data are absent. Also, some conditions for in vitro phosphatase assay on synthetic phosphopeptides are missing.

Apart from these general comments, the article is a significant step forward for our comprehension about how PP2A B56 picks its substrates. It also provides a comprehensive list of PP2A B56 and PP2A B55 targets, which is an appreciated resource for the field. For these reasons, I think this paper should be considered for publication in EMBO Journal after the following points have been addressed:

Major concerns:

1- B56 mass spectrometry strategy is extensively document, which it is not the case for B55 mass spectrometry (with the use of Arpp19 inhibitor). To be really able to compare the different phosphatase substrates, it is necessary to include data equivalent to Figure 1C-D and 2A-B but in the context of the B55 inhibitor. It will give an overview of the cell cycle phenotype and a global visualisation of the mass spectrometry data (not only logo motif).

Then, it would be interesting to know, in the common phosphosites identified, the overlap of phosphosites up-regulated in absence of B55 or B56. I am not sure if the 2% of overlap indicated p8 line 6 takes only into consideration the common phosphosites quantified in both datasets.

2- The fact that the regulatory subunit modulates the catalytic subunit to influence the phosphosite preference is, in my opinion, the most captivating part of the article. The authors use synthetic phosphopeptides with the phosphosite surrounded by different amino acid sequences. The results obtained for PP2A B55 are very interesting and convincing. In my opinion, Figure S2C would deserve to be part of the main figures. However, the authors do not demonstrate in vitro PP2A B56 preference for basophilic residues upstream of the phosphosite. PP2A B55 preferentially dephosphorylates threonine so using threonine as phosphoresidue makes sense but PP2A B56 does not show this preference. For this reason, the same kind of analysis needs to be performed by using serine phosphoresidues in presence of PP2A B55 or PP2A B56. This is an important experiment missing to have a complete story.

3- Similarly to point 2, I am confused by the choice of Cdc20 48-78 fragment to address in vitro the importance of the docking motif position. This sequence contains 3TP sites, which are dephosphorylated by PP2A B55 (Hein et al., 2017) and one potential RxxS site (S51) is also present. However, the authors identified in their screen 4 phosphosites dephosphorylated by PP2A B56 (S134, S153, T157, S160), which are not included in the fragment analysed.

Why did the authors pick this fragment, which is enriched in TP phosphosites and not appropriate for a PP2A B56-dependent dephosphorylation? Do they think that the serine S51 is the one dephosphorylated after addition of the LxxIxE motif? Or do they think that the addition of LxxIxE motif can favour the dephosphorylation of TP sites, which are normally PP2A B55 targets and not PP2A B56 targets?

Also, can the authors clarify which phosphosite is the starting point for the 12aa, 70aa and 130aa constructs (T55? T70?...)

4- Concerning the Fox03 experiment, the authors use a phospho-antibody recognising LxRxxpS/pT sequence (Figure 3D). In the text, it is mentioned that the antibody recognises pT32/pS253. Do the author have any data supporting that the antibody recognises precisely these phosphosites and not other possible phosphosites with the same sequence as S75, S315, S413 or S551?

Also, out of curiosity, did the authors try to move the docking site upstream of the phosphosites to see if the orientation makes a difference?

5- It is a pity that the authors do not discuss at all the results obtained for ADAM17, which is one of the main messages of the paper according to the abstract.

Minor concerns:

6- I am wondering if the B56 inhibitor used in this article is the same (or an upgraded version) of the one used in Kruse and al., 2018. If this is the case it should be clearly mentioned.

7- The results indicate that more phosphosites are up-regulated in G1/S than in mitosis in a B56 inhibitor context. It would be nice to have a comment about it. Is PP2A Rts1 activity variable during the cell cycle?

8- Figure 1E legend should be extended to redefine precisely what the authors call B56 SLiM motif and a colour code explanation is missing. The dots observed are either blue or pink suggesting no overlap between the proteins "B56 SLiM" or "B56 interactor", which is counterintuitive. We can observe the overlap in Table S2 but it will be nice to visualise it quickly on the figure.

Figure 2A and B, the pink dots defining a B56 SLiM should also be visible in the black dots (not significantly increased).

9- On Figure 2C-E and S2B, it would help the reader if the authors could write directly on the figure the category and number of phosphosites on the top and on the bottom of the logo motifs.

10- I am not sure what Figure 2F shows, the figure legend should be extended. Does that mean that around 70% of phosphosites carrying a (R/K)(R/K)xp(S/T) sequence are up-regulated in presence of B56 inhibitor in G1/S or 70% of the phosphosites up-regulated carry a (R/K)(R/K)Xp(S/T) sequence ? Also, in Table S3, the majority of phosphosites up-regulated seems to be on (R/K)XXp(S/T) sequences.

11- PP2A B56's specificity for basophilic residues in -3-2 position and the deselection for proline residues in +1 is very clear in the logo motifs Figure 2C-D and S2A-B. But the authors claim that they observe a preference for acidophilic residues upstream of the phosphosite (in -2) suggesting that PP2A B56 might dephosphorylate Plk1 residues (p7 -line 6, p14 line 16). The -2 acidophilic preference is observed in only one of the four logo motifs and this is quite counterintuitive with the -2 basophilic preference. Because the "E" enrichment is very low and observed only once (Figure 2D), I don't think the authors can make such a claim.

12- The word "regulated" is used many times in the article and means either "decrease or increase of phosphorylation level". This is confusing, especially in logo motifs where a regulated phosphosite can be either a higher level of phosphorylation (up-regulated) or lower level (down-regulated) (Figure S1D). It is also the case in the different excel sheets where "regulated" is used in Table S5 and S6.

13- Figure legend S1 : "each circle represents...red line" should be in part C, not B.

- Figure S1C, the meaning of "Ctr" and "wt" under the graph is not totally clear.

- Figure S2-E, there are two typos : PP2A-B56 and Cdc20 49-78 (while in the text it says 48-78)

- p31 line 9, "missing value imputed were imputed" is written twice. Also, it is not clear about which missing value the authors are referring to and the strategy of imputation should be detailed.

14- It would be great if the authors deposit their MS raw data on a proteomic platform to make them more accessible.

Referee #2:

General summary:

In this manuscript, Kruse et al. use phosphoproteomics techniques, combined with other validation strategies, to identify substrates and substrate specificity determinants of the B56 class of PP2A holoenzymes. The approach is elegant, and makes use of a 'B56-specific inhibitor', consisting of 4 copies of a previously identified high affinity B56-binding motif (LxxIxE) (by the authors) that is expressed in HeLa cells. Subsequent phosphoproteomics analysis was done in G1/S or M synchronized cells, and identified 900 differentially phosphorylated proteins, 42 of which contained an LxxIxE motif, and 667 of which were proven interactors of an LxxIxE motif-containing protein. Analysis of the phospho-site context revealed a strong preference for basic AAs upstream of the phospho-site in G1/S and basic or acidic sites in M; in both cases, a Pro residue at the +1 position was disfavored. This was further validated by *in vitro* dephosphorylation (by PP2A-B56 α) of phospho-peptides purified from cells, and by adding an LxxIxE inhibitor peptide to cell-free extracts and determination of the decrease in dephosphorylation. Authors also provide a comprehensive dataset on potential PP2A-B55 substrates by using a B55-specific inhibitor (thio-phosphorylated Arpp19) in cell-free extracts. Comparison with the PP2A-B56 dataset revealed striking differences in the preferred context of the phospho-site of both holoenzyme classes, especially downstream of the P-site, where B55 favors a Pro at +1, and basic AAs further downstream. This was further validated in *in vitro* PPase assays.

Next, the influence on dephosphorylation of the position of the LxxIxE motif relative to the dephosphorylation site was assessed, as well as the influence of the binding affinity of the phosphatase to the LxxIxE motif - using FoxO3 and cdc20 as examples. Both parameters appeared important for efficient dephosphorylation. In addition, upon inactivation of the original LxxIxE motif in the substrate, and addition of another higher-affinity binding LxxIxE motif at another location in the substrate, dephosphorylation could be sustained, arguing against a strict key-in-lock model determining substrate dephosphorylation.

In a final set of experiments, the authors validate ADAM17, an LxxIxE-containing protein coming out of their PP2A-B56 substrate screen, as a novel cellular target of PP2A-B56 tumor suppressor activity. By engineering the original LxxIxE motif of ADAM17 into a higher-affinity one, they demonstrate the functional importance of increased PP2A-B56 binding to decrease (oncogenic) ADAM17 shedding activity, to decrease proliferation and invasion potential of a colon cancer cell line, and to decrease breast tumor growth in mice. However, whether this increased affinity for PP2A-B56 has a corresponding effect on ADAM17 (de)phosphorylation state, was not assessed. Therefore, it remains unclear whether the phosphorylation of ADAM17 itself, or potentially of an ADAM17 interacting protein, is affected by PP2A-B56, and thereby mediates the tumor suppressing ability of PP2A-B56.

The manuscript of Kruse et al. clearly provides important novel insights into how multi-subunit phosphatases such as PP2A can achieve substrate specificity. The elaborate and nicely presented phosphoproteomics datasets are moreover valuable resources for future follow-up studies. Like this, the manuscript is not just of high importance to the PP2A field, but certainly also addresses the broader area of reversible protein phosphorylation-regulated cell signaling, and should therefore be of general interest.

Major concerns:

I have no major concerns regarding the first two parts of the manuscript.

My main concerns pertain to the last part, in which ADAM17 is proposed as a new potential substrate of PP2A-B56 tumor suppressor activity. Although modulation (=increase) of three ADAM17 phosphorylation sites by expression of the B56 inhibitor probe in cells is shown (Fig 4D), no data are presented on the modulation of these sites in the ADAM17 variants with altered LxxIxE motifs. Is the phosphorylation indeed increased in the non-PP2A-B56 binding I761A variant? And is the phosphorylation indeed decreased in the higher-affinity binding LEE variant? If so, although perhaps in part published (?), how would shedding activity, proliferation, invasion and tumor growth *in vivo* be affected in reconstituted ADAM17^{-/-} cell lines with non-phosphorylatable or phospho-mimicking ADAM17 mutants? If phosphorylation of ADAM17 I761A or LEE is not affected, which ADAM17 binding partner might be regulated by PP2A-B56 and thus mediate the observed functional effects of increased PP2A-B56 binding to the ADAM17 LEE mutant?

Additional minor suggestions for improvement of data presentation, analysis and/or writing:

-abstract: line 7 and 13: suggest to change 'phosphorylation site' into 'dephosphorylation site', as also

written in the title.

-page 4, 5: specify the precise sequence of the 4x LxxIxE motif used in ITC experiments, and used for transfection in the context of YFP-fusion protein (can be added to Mat & Meth).

-the B56 family of PP2A subunits consists of at least 5 isoforms (splice variants not included).

Whenever appropriate, it would be good to specify, which isoform was used or probed for in each experiment. After all, we might not be able, at this point, to exclude isoform-specific differences in the experimental outcomes. Although in some experiments the B56 isoform used is indeed specified, this is not consistently done throughout. Same remark for B55. Please adapt in main text, figures, fig legends and Mat & Meth wherever appropriate.

-to make the potential distinction between direct and indirect PP2A-B56 substrates, would it make sense to compare the overlaps (displayed as Venn-diagrams with numbers) between the hits from Table S3 with those of Table S4, and from Table S5 with those of Table S4?

-Figure 4G: statistics are missing - was the experiment done only once?

-discussion: line 8: suggest to change 'PP2A-B56 has limited activity towards Cdk1' into '...limited activity towards Cdk1 substrates'

-Some typos or additions in the Mat & Meth section:

Page 28: how was the recombinant B56alpha made for the ITC experiments?

Page 31, line 9: remove once 'missing values were imputed'

Page 34: what was the source of the purified MASTL/Gwl kinase?

Page 34, line 15: remove '650 rpm'; line 18: snap-frozen (?) (same on p 35 line 5)

Page 35, line 11: delete 'as processes'

Page 36: specify concentration of calyculin A used

Page 40, line 19: typo BALB/c mice

Referee #3:

This manuscript by Kruse et al. sets about to understand and define substrate recognition of the protein phosphatase catalytic subunit PP2A through the regulatory subunit B56. Overall the data appears valid, in that there appears to be B56-mediated regulation of substrate preference for PP2A, in terms of the motif that is dephosphorylated, likely through the presence of a concurrent B56-specific binding motif. However, critically they do not consider the phosphosite localisation confidence of their phosphoproteomics data at any stage, and thus are using low confident phosphosites in all their analyses (please see below). Inclusion of these low confidence phosphosites will undoubtedly be biasing their results and needs to be considered. Some of the conclusions e.g. with respect to EGFR signalling, are also slightly questionable based on the data presented. The manuscript was quite dense to read, and more clarity generally in the writing and the figure legends would assist the reader in following the story. Figures should be generally understandable with the legend alone, in the absence of reading the manuscript text, and this was not always the case. Oftentimes, it was also hard to follow the reasoning behind the experiments that were being presented - the rationale came though eventually, but it would have been better if the purpose of a given set of experiments was explained at the outset.

The abstract makes a general statement that B-subunits directly affect the phosphorylation site preferences of the PP2a catalytic subunit - while this may be true, this paper focusses solely on demonstrating B56-directed preference (and discrimination from B55). The abstract should therefore state this and refrain from the more general (and yet unverified) inference of all regulatory subunits.

P3, line 13 - please be specific about what type of cells you are referring to here (eukaryotic? Mammalian? All including prokaryotic systems)?

P4, l14 - a reference is needed to support the statement of the role of B56 as a tumour suppressor. It would also be useful in the introduction if the authors could state briefly how the LxxIxE motif for B56 binding was identified to help people outside of the field - this is quite critical to the rest of the manuscript and really drives how they undertook their investigations.

P5, l9 - this experiment is not clearly explained - please state what eluates you are referring to.

P5, l15-17 - it is unclear to me how you evaluated/determined specificity and concluded that the B56-inhibitor is only targeting B56, as these pull down experiments also identified B65 (from SUPP T1) - this comment is also relevant to the concluding statement of p6, line 4. There is actually no way of determining (from these particular experiments) if these binding partners were direct or indirect, so it is unclear to me how the authors came to the conclusion that these proteins are direct

interactors.

P5, 120 - again, it is assumed that the reader knows that the purpose of the experiment is (i.e. to identify YFO-B56 binding partners). It would be really helpful to add an extra line to state this explicitly.

The timing of the synchronisation and B56 protein induction are a little confusing and seem to differ between the text, the image schematic, and the methods. Please could you check? It would be useful to state in the main text at what point B56 protein expressions was induced relative to the timing of the cell stage synchronisation.

P6 122-23 - it is unclear to me what the authors mean by 'dephosphorylation of a site was a specific event and did not correlate with other sites on that protein also being dephosphorylated' - is it purely that the not all sites were dephosphorylated? Did the authors actually quantify levels (stoichiometry) of phosphorylation of all sites and undertake a proper correlation analysis?

Fig 2C - why is there nothing at position -1 or 0 in these IceLogo plots?

Fig 2F - the legend on the actual figure is difficult to understand due to the spacing between the text and the grey bars. More detail is also needed in the actual figure legend to make it easier to understand.

Pg8, 117 - how were the sequences of these peptides selected - they do not match directly with the enriched residues as described in fig2 C-E. In my opinion, it would have been more powerful to demonstrate consensus requirements if you had started with the same basic sequence and made variations at a single sites e.g. Pro at +1, R/K at -2/-2 (similar to that reported in the supplementary data). As it is you have a poorly defined sequence, and the absolute requirements of specific residues have not been demonstrated. Have the authors evaluated the ability to dephosphorylate pSer?

There is no time course data from the peptide panel presented in supplementary data - I think it would be have been more useful to have the time-course data for (variations of) these peptides in the main manuscript as an extended dataset, as it explores the limits in vitro of the motif that you have defined from the phosphoproteomics experiment.

P10, line 6 - the authors discuss using an antibody to evaluate "pY32/pS253". An antibody against pS253 is not mentioned in the methods that I can see, and I am finding it hard to understand how this experiment was performed. I have come to the conclusion that this antibody cross-reacts with both sites (?) This needs some explanation. Otherwise, it would make more sense to look at these sites individually. Why were all 3 sites (including pS413) not looked at in both the total lysate and the pull-downs?

P10, line 15 - the authors state the "phosphorylation of FoxO3 at T32, S253 and S2644 promotes its retention in the cytoplasm". I am not convinced that the authors have actually demonstrated this as the WT data do not seem to support this statement. What they appear to have is some correlation, they have not yet demonstrated to my mind that phosphorylation at one of all of these sites is a causative factor in subcellular localisation. Demonstration of this would require analysis of subcellular localisation upon mutation (CRISPR/Cas9) or introduction of the mutations (phosphomimetic and/or phosphonull) in a FoxO3 depleted background system.

P11, line 5 - "in vitro engineering" of what? Please clarify.

P11, line 22/23 - the authors refer to fold-change of a phosphorylation site, but it would be good to remind the reviewer/reader at this stage what conditions elicited this fold change.

P13 - there is some nice data showing the effect of the ADAM17 domain variants in cell-based proliferation and invasion - it would also be interesting to map the phosphosites these ADAM17 variants to show that there is a quantitative change in the expected PP2A-regulated sites upon disruption (o enhancement) of its ability to be bound by B56.

Based on the data presented (and the overexposed western blot), I am currently not convinced of the data presented in Fig 4G regarding the effect on EGFR phosphorylation (and thus EGFR signalling) and worry that this might be over-interpreted.

The western blot in Fig 4C, and the decreased electrophoretic mobility of PP2A-A suggests that there may be a specific form (modified?) of this protein that binds B56. Have the authors looked at whether there are specific modifications on this protein - how do they explain this band shift?

Methods

Critically, at no point in the manuscript do the authors mention how they filter their phosphoproteomics data for phosphosite localisation confidence, or even how/if they consider phosphosite localisation confidence - this is obviously important as they start to make predictions about substrate recognition. This is a critical omission that needs to be addressed. Although site localisation confidence appears in sup table 3 (column M) - they do not appear to do anything with this information, and about 40% of the data in the first datasheet have localisation scores below 0.75

which should be stripped from all subsequent analyses that consider site specificity.

It is also extremely important that they make all their primary and search MS data available (e.g. be deposition in PRIDE/ProteomeXchange) so that it can be searched. I'm not sure if it would currently be possible to follow all the methods and repeat their studies, so some additional information throughout would be useful. E.g.:

p28, line 10 - what cells?,

p28, line 13 - how were "peaks" detected (presumably UV, what wavelength?), what flow rate/gradient was used, what was the buffer composition?

Pg 30, line 2-3 - please specify the amount of buffer used for cell lysis, and washing etc.

It is somewhat confusing to be discussing the LC-MS analysis before presenting how the samples were prepared and the peptides generated.

Possibly for reviewers/editor information only, but it would be useful to explain why you are using a database that is 6 years old - this is not typical, but I appreciate that it may have taken that long to complete the study (?)

Please can you clarify how the normalisation was performed for quantification. As the TMT labelling was done post-phosphopeptide enrichment, normalisation between conditions will be affected by the total phosphopeptide content (and efficiency of phosphopeptide enrichment). Thus it is unclear how you can adequately normalise to define fold change between samples in this manner, particularly as you know that you are disrupting the efficiency of PP2A target binding.

P35, lines 14 - please include details of how the off-line separation was performed of the TMT-labelled samples.

P35, line 18 - 8% is a relatively high starting MeCN concentration for peptide elution from C18. Do you think this may be biasing your cohort of identified phosphopeptide motifs given that you will likely not be seeing any of the really hydrophilic peptides? I would be interested to see how this changes from ~3% MeCN.

P35, line 22 - please state how much calyculin A (activity units) were used - what were the reaction conditions/buffer?

Referee #1:

Kruse and al. investigate how PP2A B56 and PP2A B55 phosphatases target their substrates for dephosphorylation. To do so, they have developed a strategy combining the use of specific phosphatase inhibitors and global mass spectrometry analysis to define which kind of phospho amino-acid sequences are recognised by the phosphatases. Their results suggest that regulatory subunits control substrate recognition at two different levels. The first interesting concept is that the regulatory subunits B56 or B55 modulate the PP2A catalytic subunit resulting in the recognition of different phosphomotifs. The second concept is based on the fact that B56 regulatory subunit recognises a docking site at the distance of the phosphosite, which contributes to substrate recognition. Using Cdc20 and FoxO3 as model substrates, their work highlights the importance of the distance between the docking site and the phosphosite for efficient dephosphorylation. By moving the docking site at different places of the protein or by increasing the quantity of docking motifs on the substrate, they also show some flexibility suggesting that the binding strength (and not only the docking position) is important to determine which phosphosites are dephosphorylated. Finally, they show the importance of the B56 docking motif on the protein ADAM17 to negatively regulate ADAM17 activity and tumour growth in mice.

Several recent publications from this lab have significantly contributed to our knowledge about phosphatase specificity. Overall, the article is well written and easy to follow. The authors present the article as a side-by-side comparison of the two PP2A holoenzymes but in reality they put much more emphasis on B56 than B55, for which some data are absent. Also, some conditions for in vitro phosphatase assay on synthetic phosphopeptides are missing.

Apart from these general comments, the article is a significant step forward for our comprehension about how PP2A B56 picks its substrates. It also provides a comprehensive list of PP2A B56 and PP2A B55 targets, which is an appreciated resource for the field. For these reasons, I think this paper should be considered for publication in EMBO Journal after the following points have been addressed:

We thank the reviewer for the suggestions to improve our manuscript.

Major concerns:

1- B56 mass spectrometry strategy is extensively document, which it is not the case for B55 mass spectrometry (with the use of Arpp19 inhibitor). To be really able to compare the different phosphatase substrates, it is necessary to include data equivalent to Figure 1C-D and 2A-B but in the context of the B55 inhibitor. It will give an overview of the cell cycle phenotype and a global visualisation of the mass spectrometry data (not only logo motif).

Our response:

We used two different experimental approaches to identify B55 and B56 substrates because of technical limitations. For B55, we used an in vitro approach where we added purified and thiophosphorylated ARPP19 WT or ARPP19 S62A to mitotic lysates.

For B56, we developed an in cell system based on the inducible expression of inhibitory SLiM sequences.

Because ARPP19 can only be thiophosphorylated in vitro, we cannot include the same kind of analysis as in Figure 1C and D as well as 2A and B. Instead we have included a validation of the B55 approach where cellular lysates are treated with either thiophosphorylated ARPP19 WT or ARPP19 S62A. The lysate samples are subjected to WB and probed with anti-pTP antibodies. New figure EV2H.

Then, it would be interesting to know, in the common phosphosites identified, the overlap of phosphosites up-regulated in absence of B55 or B56. I am not sure if the 2% of overlap indicated p8 line 6 takes only into consideration the common phosphosites quantified in both datasets.

Our response:

We adjusted the sentence below to indicate that the comparison is made only with regulated sites.

This identified 1405 PP2A-B55 regulated sites (\log_2 ratio > 0.8 (1.75-fold), p-value < 0.05) of which less than 1.3% were shared with the regulated sites identified in the PP2A-B56 data sets

We further adjusted this to only include localized sites.

2- The fact that the regulatory subunit modulates the catalytic subunit to influence the phosphosite preference is, in my opinion, the most captivating part of the article. The authors use synthetic phosphopeptides with the phosphosite surrounded by different amino acid sequences. The results obtained for PP2A B55 are very interesting and convincing. In my opinion, Figure S2C would deserve to be part of the main figures.

Our response:

We agree and have done this in the revised manuscript. New figure 3B.

However, the authors do not demonstrate in vitro PP2A B56 preference for basophilic residues upstream of the phosphosite.

Our response:

We agree and have included the analysis of series of phosphopeptides to address this point in the revised manuscript (Figure 3C).

In vitro, the deselection of proline in the +1 position was confirmed, whereas we did not observe an inherent preference of PP2A-B56 for basophilic residues upstream of the phosphosite. We have adjusted our claims accordingly.

PP2A B55 preferentially dephosphorylates threonine so using threonine as phosphoresidue makes sense but PP2A B56 does not show this preference. For this reason, the same kind of analysis needs to be performed by using serine phosphoresidues in presence of PP2A B55 or PP2A B56. This is an important experiment missing to have a complete story.

Our response:

We have included a comparison of phosphothreonine and phosphoserine model peptides and find that PP2A-B56 dephosphorylates these equally well, consistent with the iceLogo for PP2A-B56 showing no specific enrichment for threonine. New figure 3D.

3- Similarly to point 2, I am confused by the choice of Cdc20 48-78 fragment to address in vitro the importance of the docking motif position. This sequence contains 3TP sites, which are dephosphorylated by PP2A B55 (Hein et al., 2017) and one potential RxxS site (S51) is also present. However, the authors identified in their screen 4 phosphosites dephosphorylated by PP2A B56 (S134, S153, T157, S160), which are not included in the fragment analysed.

Why did the authors pick this fragment, which is enriched in TP phosphosites and not appropriate for a PP2A B56-dependent dephosphorylation?

Our response:

We are well aware that picking a Cdc20 fragment with three TP sites for the in vitro analysis of PP2A-B56 phosphatase activity may seem counterintuitive, but also want to point out that a few TP/SP sites are detected in our proteomic screens. So while PP2A-B56 clearly dephosphorylates TP/SP sites poorly, it can do it. In this particular case, where the objective is to investigate effects of the LxxIxE motif positioning and affinity (and not the catalytic preference of PP2A-B56), we think it is scientifically justified to apply substrates with phosphorylation sites sub-optimal for PP2A-B56. We have previously used this GST Cdc20 fragment for engineering both for PP2A-B55 (Hein et al 2017) and PP4 (Ueki et al 2019) and it is easy to purify and phosphorylate, which is why we prefer to use it for engineering experiments.

We also want to point out that a recent paper from the Yamano lab (Fujimitsu and Yamano, EMBO Reports 2020) shows that binding of PP2A-B56 to an LxxIxE motif in APC1 leads to dephosphorylation of these TP sites in Cdc20, justifying our use of Cdc20 as a model substrate. We have now cited this paper that appeared during revision.

Do they think that the serine S51 is the one dephosphorylated after addition of the LxxIxE motif ?

Our response:

We know from previous work that S51 is not being phosphorylated on the GST-Cdc20 49-78 fragment when we use cdk1 as the kinase. Which is the kinase used here. We know this because the GST-cdc20 fragment where the three TP sites are mutated to alanine show no phosphorylation when treated with cdk1 (Hein et al 2017).

Or do they think that the addition of LxxIxE motif can favour the dephosphorylation of TP sites, which are normally PP2A B55 targets and not PP2A B56 targets ?

Our response:

This indeed seems to be the case since in the GST-cdc20 fragment, containing a mutated LxxIxE motif (AxxAxA), virtually no dephosphorylation by PP2A-B56 is observed. This would also be consistent with

the recent work from the Yamano lab where an LxxIxE motif on APC1 likely brings PP2A-B56 in proximity of Cdc20.

Also, can the authors clarify which phosphosite is the starting point for the 12aa, 70aa and 130aa constructs (T55? T70?...)

Our response:

T70 is the starting point. We have clarified this in the revised manuscript. Figure 4F.

4- Concerning the FoxO3 experiment, the authors use a phospho-antibody recognising LxRxxpS/pT sequence (Figure 3D). In the text, it is mentioned that the antibody recognises pT32/pS253. Do the author have any data supporting that the antibody recognises precisely these phosphosites and not other possible phosphosites with the same sequence as S75, S315, S413 or S551?

Our response:

We agree that using a phospho-antibody recognizing an LxRxxpS/pT sequence as a surrogate for T32 and S253 phosphorylation is not optimal. We tested extensively a number of commercially available pT32 or pS253 antibodies. Neither of these antibodies worked in our hands. Fortunately, we managed to produce a pS253 phospho-antibody in-house, which became available during the revision period. We have repeated experiments with this antibody and included these results in the revised manuscript instead of the data obtained with the LxRxxpS/pT antibody. Figure 4D. The data are fully consistent with the data in the original submission.

Also, out of curiosity, did the authors try to move the docking site upstream of the phosphosites to see if the orientation makes a difference?

Our response:

This is an interesting suggestion, but unfortunately we have not explored this.

5- It is a pity that the authors do not discuss at all the results obtained for ADAM17, which is one of the main messages of the paper according to the abstract.

Our response:

We agree with the reviewer and have now included a discussion of the ADAM17 results in the revised manuscript.

Minor concerns:

6- I am wondering if the B56 inhibitor used in this article is the same (or an upgraded version) of the one used in Kruse and al., 2018. If this is the case it should be clearly mentioned.

Our response:

It is. We have now stated that in the revised manuscript and also refer to Kruse et al., 2018.

7- The results indicate that more phosphosites are up-regulated in G1/S than in mitosis in a B56 inhibitor context. It would be nice to have a comment about it. Is PP2A Rts1 activity variable during the cell cycle?

Our response:

That is correct. However, we do not know whether there is an obvious explanation for this but possibly it can be due to the high occupancy of phosphorylation sites in mitosis which would prevent us from detecting an increase upon PP2A inhibition.

To the best of our knowledge PP2A-B56 is constitutively active throughout the cell cycle but this has not been explored extensively.

8- Figure 1E legend should be extended to redefine precisely what the authors call B56 SLiM motif and a colour code explanation is missing. The dots observed are either blue or pink suggesting no overlap between the proteins "B56 SLiM" or "B56 interactor", which is counterintuitive. We can observe the overlap in Table S2 but it will be nice to visualise it quickly on the figure.

Figure 2A and B, the pink dots defining a B56 SLiM should also be visible in the black dots (not significantly increased).

Our response:

We have added a description of the color code to the legend of Figure 1E.

9- On Figure 2C-E and S2B, it would help the reader if the authors could write directly on the figure the category and number of phosphosites on the top and on the bottom of the logo motifs.

Our response:

We have added the number of single localized phosphorylation sites that were used to generate the icelogs to the indicated Figures.

10- I am not sure what Figure 2F shows, the figure legend should be extended. Does that mean that around 70% of phosphosites carrying a (R/K)(R/K)Xp(S/T) sequence are up-regulated in presence of B56 inhibitor in G1/S or 70% of the phosphosites up-regulated carry a (R/K)(R/K)Xp(S/T) sequence ? Also, in Table S3, the majority of phosphosites up-regulated seems to be on (R/K)XXp(S/T) sequences.

Our response:

We apologize for the confusion and have added additional text to the legend for clarification and a more transparent figure 2F. Briefly, we determined the distribution of the three types of phosphorylation consensus motifs in the up-regulated phosphorylation sites. The reviewer has identified correctly that the majority of phosphorylation sites that increased upon B56 inhibition have a basophilic phosphorylation site motif.

11- PP2A B56's specificity for basophilic residues in -3-2 position and the deselection for proline residues in +1 is very clear in the logo motifs Figure 2C-D and S2A-B. But the authors claim that they observe a preference for acidophilic residues upstream of the phosphosite (in -2) suggesting that PP2A B56 might dephosphorylate Plk1 residues (p7 -line 6, p14 line 16). The -2 acidophilic preference is observed in only one of the four logo motifs and this is quite counterintuitive with the -2 basophilic preference. Because the "E" enrichment is very low and observed only once (Figure 2D), I don't think the authors can make such a claim.

Our response:

We agree with the reviewer and have adjusted the description in the manuscript.

12- The word "regulated" is used many times in the article and means either "decrease or increase of phosphorylation level". This is confusing, especially in logo motifs where a regulated phosphosite can be either a higher level of phosphorylation (up-regulated) or lower level (down-regulated) (Figure S1D). It is also the case in the different excel sheets where "regulated" is used in Table S5 and S6.

Our response:

We have specified throughout the manuscript whether regulated sites are up-regulated or down-regulated. With respect to the different excel sheets, we use the term "regulated" because here both up-regulated and down-regulated phosphorylation sites are present.

13- Figure legend S1 : "each circle represents...red line" should be in part C, not B.

- Figure S1C, the meaning of "Ctr" and "wt" under the graph is not totally clear.

- Figure S2-E, there are two typos : PP22A-B56 and Cdc20 49-78 (while in the text it says 48-78)

- p31 line 9, "missing value imputed were imputed" is written twice. Also, it is not clear about which missing value the authors are referring to and the strategy of imputation should be detailed.

Our response:

Typos have been corrected in text and figures as suggested.

14- It would be great if the authors deposit their MS raw data on a proteomic platform to make them more accessible.

Our response:

We have deposited the data to ProteomeXchange PXD015205, MassIVE MSV000084245, password p730 and indicated this in the acknowledgment section of the manuscript.

Referee #2:

General summary:

In this manuscript, Kruse et al. use phosphoproteomics techniques, combined with other validation strategies, to identify substrates and substrate specificity determinants of the B56 class of PP2A holoenzymes. The approach is elegant, and makes use of a 'B56-specific inhibitor', consisting of 4 copies of a previously identified high affinity B56-binding motif (LxxIxE) (by the authors) that is expressed in HeLa cells. Subsequent phosphoproteomics analysis was done in G1/S or M synchronized cells, and identified 900 differentially phosphorylated proteins, 42 of which contained an LxxIxE motif, and 667 of which were proven interactors of an LxxIxE motif-containing protein. Analysis of the phospho-site context revealed a strong preference for basic AAs upstream of the phospho-site in G1/S and basic or acidic sites in M; in both cases, a Pro residue at the +1 position was disfavored. This was further validated by in vitro dephosphorylation (by PP2A-B56alpha) of phospho-peptides purified from cells, and by adding an LxxIxE inhibitor peptide to cell-free extracts and determination of the decrease in dephosphorylation. Authors also provide a comprehensive dataset on potential PP2A-B55 substrates by using a B55-specific inhibitor (thio-phosphorylated Arpp19) in cell-free extracts. Comparison with the PP2A-B56 dataset revealed striking differences in the preferred context of the phospho-site of both holoenzyme classes, especially downstream of the P-site, where B55 favors a Pro at +1, and basic AAs further downstream. This was further validated in in vitro PPase assays.

Next, the influence on dephosphorylation of the position of the LxxIxE motif relative to the dephosphorylation site was assessed, as well as the influence of the binding affinity of the phosphatase to the LxxIxE motif - using FoxO3 and cdc20 as examples. Both parameters appeared important for efficient dephosphorylation. In addition, upon inactivation of the original LxxIxE motif in the substrate, and addition of another higher-affinity binding LxxIxE motif at another location in the substrate, dephosphorylation could be sustained, arguing against a strict key-in-lock model determining substrate dephosphorylation.

In a final set of experiments, the authors validate ADAM17, an LxxIxE-containing protein coming out of their PP2A-B56 substrate screen, as a novel cellular target of PP2A-B56 tumor suppressor activity. By engineering the original LxxIxE motif of ADAM17 into a higher-affinity one, they demonstrate the functional importance of increased PP2A-B56 binding to decrease (oncogenic) ADAM17 shedding activity, to decrease proliferation and invasion potential of a colon cancer cell line, and to decrease breast tumor growth in mice. However, whether this increased affinity for PP2A-B56 has a corresponding effect on ADAM17 (de)phosphorylation state, was not assessed. Therefore, it remains unclear whether the phosphorylation of ADAM17 itself, or potentially of an ADAM17 interacting protein, is affected by PP2A-B56, and thereby mediates the tumor suppressing ability of PP2A-B56.

The manuscript of Kruse et al. clearly provides important novel insights into how multi-subunit phosphatases such as PP2A can achieve substrate specificity. The elaborate and nicely presented phosphoproteomics datasets are moreover valuable resources for future follow-up studies. Like this, the manuscript is not just of high importance to the PP2A field, but certainly also addresses the broader area of reversible protein phosphorylation-regulated cell signaling, and should therefore be of general interest.

We thank the reviewer for the suggestions to improve our manuscript.

Major concerns:

I have no major concerns regarding the first two parts of the manuscript.

My main concerns pertain to the last part, in which ADAM17 is proposed as a new potential substrate of

PP2A-B56 tumor suppressor activity. Although modulation (=increase) of three ADAM17 phosphorylation sites by expression of the B56 inhibitor probe in cells is shown (Fig 4D), no data are presented on the modulation of these sites in the ADAM17 variants with altered LxxIxE motifs. Is the phosphorylation indeed increased in the non-PP2A-B56 binding I761A variant? And is the phosphorylation indeed decreased in the higher-affinity binding LEE variant? If so, although perhaps in part published (?),

Our response:

These are indeed relevant questions. We did provide a comparison of the phosphorylation status of immunopurified ADAM17 wt and the I762A variant, using quantitative mass spectrometry, in the first submission (Fig 4D in original version). This showed that phosphorylation of T735 and S808 is increased in the I762A variant.

We apologize that this was not very clearly written/shown in the manuscript and Fig 4D. We have clarified this in the revised manuscript and also included a more transparent figure. New figure 5G.

Furthermore, we have now expanded on this by performing in vitro dephosphorylation assays using GST tagged C-terminal fragments of ADAM17 WT, I762A and LEE variants. These fragments were phosphorylated with PKA and subsequently, we followed dephosphorylation kinetics upon addition of PP2A-B56. The data revealed that the LxxIxE motif stimulated dephosphorylation of ADAM17 in vitro. New figure 5D.

Collectively, we find that these data support the idea that at least one substrate of ADAM17 bound PP2A-B56 is ADAM17 itself.

how would shedding activity, proliferation, invasion and tumor growth in vivo be affected in reconstituted ADAM17^{-/-} cell lines with non-phosphorylatable or phospho-mimicking ADAM17 mutants?

Our response:

We did test this on the T735 site identified in the PP2A-B56 phosphoproteomics screen. Neither the T735A nor the T735D ADAM17 mutant gave rise to significant phenotypes. This is not surprising since PP2A-B56 seems to be working on several ADAM17 phosphorylation sites and likely also on sites on ADAM17 binding partners.

So the phenotypes observed when uncoupling (or increasing) the binding between PP2A-B56 and ADAM17 is probably a combined effect of several regulated phosphorylation sites.

We have commented on this in the revised manuscript.

If phosphorylation of ADAM17 I761A or LEE is not affected, which ADAM17 binding partner might be regulated by PP2A-B56 and thus mediate the observed functional effects of increased PP2A-B56 binding to the ADAM17 LEE mutant?

Our response:

Several previous publications and data presented here have shown that PP2A-B56 also works on phosphorylation sites on binding partners of proteins containing LxxIxE motifs. We find it highly likely that this is also true for binding partners of ADAM17. Potential candidates are iRhom1 and 2, as well as PACS-2, whose functions are heavily phospho-regulated.

Additional minor suggestions for improvement of data presentation, analysis and/or writing:
-abstract: line 7 and 13: suggest to change 'phosphorylation site' into 'dephosphorylation site', as also written in the title.

Our response:

We have adjusted the text as suggested by the reviewer.

-page 4, 5: specify the precise sequence of the 4x LxxIxE motif used in ITC experiments, and used for transfection in the context of YFP-fusion protein (can be added to Mat & Meth).

Our response:

Sequences have been added to Mat & Meth as suggested.

-the B56 family of PP2A subunits consists of at least 5 isoforms (splice variants not included). Whenever appropriate, it would be good to specify, which isoform was used or probed for in each experiment. After all, we might not be able, at this point, to exclude isoform-specific differences in the experimental outcomes. Although in some experiments the B56 isoform used is indeed specified, this is not consistently done throughout. Same remark for B55. Please adapt in main text, figures, fig legends and Mat & Meth wherever appropriate.

Our response:

We have specified this wherever appropriate in the revised manuscript.

-to make the potential distinction between direct and indirect PP2A-B56 substrates, would it make sense to compare the overlaps (displayed as Venn-diagrams with numbers) between the hits from Table S3 with those of Table S4, and from Table S5 with those of Table S4?

Our response:

We appreciate the reviewer's comment. Unfortunately, both in vitro datasets are much smaller than the in cell datasets. Accordingly, we identify only 28 phosphorylation sites common between the datasets. We believe that to make the distinction between direct and indirect PP2A-B56 substrates would require additional biochemical testing. Nevertheless, it is worth noting that ADAM17 Threonine 735 is one of the 28 phosphorylation sites common between the datasets.

-Figure 4G: statistics are missing - was the experiment done only once?

Our response:

Shown was a representative WB and quantification of two independent experiments. We have repeated this experiment three more times and included proper statistics in the revised manuscript. New figure 6A.

-discussion: line 8: suggest to change 'PP2A-B56 has limited activity towards Cdk1' into '....limited activity towards Cdk1 substrates'

Our response:

Agree. We have changed this as suggested.

-Some typos or additions in the Mat & Meth section:

Page 28: how was the recombinant B56alpha made for the ITC experiments?

Our response:

As described previously (Hertz et al., 2016). We have included this reference.

Page 31, line 9: remove once 'missing values were imputed'

Our response:

Done

Page 34: what was the source of the purified MASTL/Gwl kinase?

Our response:

As described previously in Hein et al., 2017. We have included this reference.

Page 34, line 15: remove '650 rpm'; line 18: snap-frozen (?) (same on p 35 line 5)

Our response:

Corrected

Page 35, line 11: delete 'as processes'

Our response:

Done

Page 36: specify concentration of calyculin A used

Our response:

Done

Page 40, line 19: typo BALB/c mice

Our response:

Corrected

Referee #3:

This manuscript by Kruse et al. sets about to understand and define substrate recognition of the protein phosphatase catalytic subunit PP2A through the regulatory subunit B56. Overall the data appears valid, in that there appears to be B56-mediated regulation of substrate preference for PP2A, in terms of the motif that is dephosphorylated, likely through the presence of a concurrent B56-specific binding motif. However, critically they do not consider the phosphosite localisation confidence of their phosphoproteomics data at any stage, and thus are using low confident phosphosites in all their analyses (please see below). Inclusion of these low confidence phosphosites will undoubtedly be biasing their results and needs to be considered.

Our response:

We thank the reviewer for the comments and suggestions for improving our manuscript.

In the revised manuscript, we have restricted all analyses to phosphorylation sites with a localization probability of 0.75 or more. Please see the more detailed comments below.

Some of the conclusions e.g. with respect to EGFR signalling, are also slightly questionable based on the data presented. The manuscript was quite dense to read, and more clarity generally in the writing and the figure legends would assist the reader in following the story. Figures should be generally understandable with the legend alone, in the absence of reading the manuscript text, and this was not always the case. Oftentimes, it was also hard to follow the reasoning behind the experiments that were being presented - the rationale came though eventually, but it would have been better if the purpose of a given set of experiments was explained at the outset.

The abstract makes a general statement that B-subunits directly affect the phosphorylation site

preferences of the PP2a catalytic subunit - while this may be true, this paper focusses solely on demonstrating B56-directed preference (and discrimination from B55). The abstract should therefore state this and refrain from the more general (and yet unverified) inference of all regulatory subunits.

Our response:

We appreciate the reviewer's comment and have adjusted the abstract to reflect the focus on B56 and B55 subunits.

P3, line 13 - please be specific about what type of cells you are referring to here (eukaryotic? Mammalian? All including prokaryotic systems)?

Our response:

We are referring to eukaryotic cells. This has been stated explicitly in the revised manuscript.

P4, l14 - a reference is needed to support the statement of the role of B56 as a tumour suppressor. It would also be useful in the introduction if the authors could state briefly how the LxxIxE motif for B56 binding was identified to help people outside of the field - this is quite critical to the rest of the manuscript and really drives how they undertook their investigations.

Our response:

We have added additional references on the role of PP2A-B56 as a tumor suppressor and a brief description of the identification of the LxxIxE motif as suggested by the reviewer.

P5, l9 - this experiment is not clearly explained - please state what eluates you are referring to.

Our response:

Done.

P5, l15-17 - it is unclear to me how you evaluated/determined specificity and concluded that the B56-inhibitor is only targeting B56, as these pull down experiments also identified B65 (from SUPP T1) - this comment is also relevant to the concluding statement of p6, line 4. There is actually no way of determining (from these particular experiments) if these binding partners were direct or indirect, so it is unclear to me how the authors came to the conclusion that these proteins are direct interactors.

Our response:

It has been documented quite extensively in a recent series of publications that LxxIxE motifs bind directly to a conserved pocket in B56 (Hertz et al 2016, Wang et al 2016). The observation that pull-downs with the YFP-LxxIxE inhibitor only revealed binding to PP2A holoenzyme components (B56

isoforms, scaffold and catalytic subunits) is a strong argument that the inhibitor is specific for PP2A-B56.

Note that B65 (from Supp T1) is the scaffolding subunit of PP2A. We would expect to identify B65 in these pull-downs together with the catalytic subunit.

P5, l20 - again, it is assumed that the reader knows that the purpose of the experiment is (i.e. to identify YFO-B56 binding partners). It would be really helpful to add an extra line to state this explicitly. The timing of the synchronisation and B56 protein induction are a little confusing and seem to differ between the text, the image schematic, and the methods. Please could you check? It would be useful to state in the main text at what point B56 protein expressions was induced relative to the timing of the cell stage synchronisation.

Our response:

We agree and have introduced an extra line that states the purpose of the experiment: "Next, we tested whether the B56 inhibitor is able to displace PP2A-B56 interactors"

P6 l22-23 - it is unclear to me what the authors mean by 'dephosphorylation of a site was a specific event and did not correlate with other sites on that protein also being dephosphorylated' - is it purely that the not all sites were dephosphorylated? Did the authors actually quantify levels (stoichiometry) of phosphorylation of all sites and undertake a proper correlation analysis?

Our response:

Correct. Our intention was to indicate that not all sites on a specific protein are regulated by PP2A-B56. As suggested by the reviewer, we have included in the revised manuscript a correlation analysis of the B56-regulated sites ($\log_2 > 0.8$, $p\text{-value} < 0.05$, localization probability score $> 75\%$) versus all other phosphorylation sites identified and quantified on the same protein. The correlation is $R = 0.1113$ suggesting that B56-dependent changes are site-specific and do not affect all phosphorylation sites on the same protein. We have added this text and a new figure (figure EV2C) to clarify this point.

Fig 2C - why is there nothing at position -1 or 0 in these IceLogo plots?

Our response:

The icelogo plot visualizes over- and underrepresented amino acids in a dataset compared to a background dataset. Lack of amino acids in a specific position means that there is no over- or underrepresentation compared to the background. In other words, the distributions of STY in the 0 position in the substrates is the same as in the background.

Fig 2F - the legend on the actual figure is difficult to understand due to the spacing between the text and the grey bars. More detail is also needed in the actual figure legend to make it easier to understand.

Our response:

We have adjusted the legend in the figure and in the actual figure legend.

Pg8, l17 - how were the sequences of these peptides selected - they do not match directly with the enriched residues as described in fig2 C-E.

Our response:

That is correct. These peptides were designed to contain consensus phosphorylation sites conforming to physiological relevant cellular kinases: Basophilic (Protein kinase C), acidophilic (PLK1) and proline directed (Cdk1). We have clarified this in the revised manuscript. We have also expanded our analysis to additional phosphopeptides containing the enriched and deselected residues. Figure 3C.

In my opinion, it would have been more powerful to demonstrate consensus requirements if you had started with the same basic sequence and made variations at a single sites e.g. Pro at +1, R/K at -2/-2 (similar to that reported in the supplementary data). As it is you have a poorly defined sequence, and the absolute requirements of specific residues have not been demonstrated.

Our response:

We agree and have included a series of phosphopeptides that probes the consensus sequence requirements arising from the PP2A-B56 phosphoproteomics experiments (similar to what was done in the supplementary data on PP2A-B55). New figure 3C.

In vitro the deselection of proline in the +1 position was confirmed, whereas we did not observe an inherent preference of PP2A-B56 for basophilic residues upstream of the phosphosite. We have adjusted our claims accordingly.

Have the authors evaluated the ability to dephosphorylate pSer?

Our response:

Yes, we have compared phosphothreonine and phosphoserine model peptides and see similar activity consistent with our PP2A-B56 iceLogo representations.

There is no time course data from the peptide panel presented in supplementary data - I think it would be have been more useful to have the time-course data for (variations of) these peptides in the main

manuscript as an extended dataset, as it explores the limits in vitro of the motif that you have defined from the phosphoproteomics experiment.

Our response:

As such, we do not disagree with the reviewer on this point. However, for the purpose of probing consensus sequence requirements on this many peptides, we believe that assays used here are scientifically appropriate. We measure dephosphorylation activity during the reaction, so our measurements are not end point assays.

We agree with the reviewer about moving the in vitro peptide dephosphorylation data into the main manuscript and have done this in the revised version. New figures 3B and C.

P10, line 6 - the authors discuss using an antibody to evaluate "pY32/pS253". An antibody against pS253 is not mentioned in the methods that I can see, and I am finding it hard to understand how this experiment was performed. I have come to the conclusion that this antibody cross-reacts with both sites (?) This needs some explanation. Otherwise, it would make more sense to look at these sites individually.

Our response:

We agree that using a phospho-antibody recognizing an LxRxxpS/pT sequence as a surrogate for T32 and S253 phosphorylation is not optimal. We tested extensively a number of commercially available pT32 or pS253 antibodies. Neither of these antibodies worked in our hands. Fortunately, we managed to produce a pS253 phospho-antibody in-house, which became available during the revision period. We have repeated experiments with this antibody and included these results in the revised manuscript instead of the data obtained with the LxRxxpS/pT antibody. The new data with the S253 antibody is fully consistent with the data generated with the LxRxxpS/pT antibody. New figure 4D.

Why were all 3 sites (including pS413) not looked at in both the total lysate and the pull-downs?

Our response:

In our experience, some phospho-antibodies work well on whole cell lysates, whereas others only work on purified/precipitated proteins. In this case, the pS413 antibody worked well on whole cell lysates, so for this antibody there was no need for pull-downs. On the other hand, both the "pT32/pS253" and our new pS253 antibody only worked after pull-down of the target protein.

P10, line 15 - the authors state the "phosphorylation of FoxO3 at T32, S253 and S2644 promotes its retention in the cytoplasm". I am not convinced that the authors have actually demonstrated this as the WT data do not seem to support this statement. What they appear to have is some correlation, they have not yet demonstrated to my mind that phosphorylation at one of all of these sites is a causative factor in subcellular localisation. Demonstration of this would require analysis of subcellular localisation upon mutation (CRISPR/Cas9) or introduction of the mutations (phosphomimetic and/or phosphonull) in a FoxO3 depleted background system.

Our response:

It was not our intention to make any claims on the contribution of these amino acid residues to FoxO3 localization as several earlier papers have documented this. We have clarified this in the revised manuscript. As the reviewer rightfully notes, our purpose with this experiment was merely to establish a meaningful correlation between the biochemical dephosphorylation data and how the different FoxO3 variant proteins localized in vivo.

P11, line 5 - "in vitro engineering" of what? Please clarify.

Our response:

We have rephrased this sentence in the revised manuscript.

P11, line 22/23 - the authors refer to fold-change of a phosphorylation site, but it would be good to remind the reviewer/reader at this stage what conditions elicited this fold change.

Our response:

We have clarified this now.

P13 - there is some nice data showing the effect of the ADAM17 domain variants in cell-based proliferation and invasion - it would also be interesting to map the phosphosites these ADAM17 variants to show that there is a quantitative change in the expected PP2A-regulated sites upon disruption (or enhancement) of its ability to be bound by B56.

Our response:

We did provide a comparison of the phosphorylation status of immunopurified ADAM17 wt and the I762A variant using quantitative mass spectrometry in the first submission (Fig 4D in original version). This showed that phosphorylation of T735 and S808 is increased in the I762 variant.

We apologize that this is not very clearly written/shown in the manuscript and Fig 4D. We have clarified this in the revised manuscript and also included a more transparent figure. New figure 5G.

Furthermore, we have now expanded on this by performing in vitro dephosphorylation assays using GST tagged C-terminal fragments of ADAM17 WT, I762A and LEE variants. These fragments were phosphorylated with PKA and subsequently, we followed dephosphorylation kinetics upon addition of PP2A-B56. The data revealed that the LxxIxE motif stimulated dephosphorylation of ADAM17 in vitro. New figure 5D.

Collectively we find that these data support the idea that at least on substrate of ADAM17 bound PP2A-B56 is ADAM17 itself.

Based on the data presented (and the overexposed western blot), I am currently not convinced of the data presented in Fig 4G regarding the effect on EGFR phosphorylation (and thus EGFR signalling) and worry that this might be over-interpreted.

Our response:

We agree with the reviewer on the EGFR signaling experiment. We have repeated this experiment three times and included proper statistics and a western blot with less exposure. Figure 6A.

The western blot in Fig 4C, and the decreased electrophoretic mobility of PP2A-A suggests that there may be a specific form (modified?) of this protein that binds B56. Have the authors looked at whether there are specific modifications on this protein - how do they explain this band shift?

Methods

Our response:

We often see slightly decreased electrophoretic mobility of proteins in whole cell lysate compared to immuno-precipitated proteins. We do not know the reason for this.

Critically, at no point in the manuscript do the authors mention how they filter their phosphoproteomics data for phosphosite localisation confidence, or even how/if they consider phosphosite localisation confidence - this is obviously important as they start to make predictions about substrate recognition. This is a critical omission that needs to be addressed. Although site localisation confidence appears in sup table 3 (column M) - they do not appear to do anything with this information, and about 40% of the data in the first datasheet have localisation scores below 0.75 which should be stripped from all subsequent analyses that consider site specificity.

Our response:

In the revised manuscript, we have restricted our list of B55- and B56-dependent phosphorylation sites to phosphopeptides identified with a phosphorylation site localization probability of 75% or larger (see Supp Table 3, 4, 5, and 6). We have repeated all analyses displayed in Figure 2C-H, EV2C-E with phosphorylation sites with a localization probability of 75% or above. We have added text to manuscript main text and the methods section to indicate this change.

It is also extremely important that they make all their primary and search MS data available (e.g. be deposition in PRIDE/ProteomeXchange) so that it can be searched.

Our response:

We have deposited the data to ProteomeXchange PXD015205, MassIVE MSV000084245, password p730 and indicated this in the acknowledgment section of the manuscript.

I'm not sure if it would currently be possible to follow all the methods and repeat their studies, so some additional information throughout would be useful. E.g.:

p28, line 10 - what cells?,

p28, line 13 - how were "peaks" detected (presumably UV, what wavelength?), what flow rate/gradient was used, what was the buffer composition?

Pg 30, line 2-3 - please specify the amount of buffer used for cell lysis, and washing etc.

Our response:

We have added more detailed information to the Materials and Methods section throughout, as suggested by the reviewer.

It is somewhat confusing to be discussing the LC-MS analysis before presenting how the samples were prepared and the peptides generated.

Our response:

We have adjusted the order of the method section.

Possibly for reviewers/editor information only, but it would be useful to explain why you are using a database that is 6 years old - this is not typical, but I appreciate that it may have taken that long to complete the study (?)

Our response:

We appreciate the reviewers comment and agree that although the annotation of the human proteome has not changed significantly over the past ten years, it is timely to update our database. As the reviewer suggested, we used an older database to be consistent across all experiments. Comparisons of the 2013 and a download of the human Uniprot database 12/20/2019 indicates only 2.15% of entries were either renamed or replaced which should not significantly impact the studies. Moving forward, we are switching to the 2019 database.

Please can you clarify how the normalisation was performed for quantification. As the TMT labelling was done post-phosphopeptide enrichment, normalisation between conditions will be affected by the total phosphopeptide content (and efficiency of phosphopeptide enrichment). Thus it is unclear how you can adequately normalise to define fold change between samples in this manner, particularly as you know that you are disrupting the efficiency of PP2A target binding.

Our response:

This is a great point and something we think a lot about and consider carefully. Normalization is first performed after lysis using protein assays to determine the total protein content of each sample. Equal amounts of protein per sample are trypsin digested and processed in parallel by phosphopeptide enrichment and TMT labeling. TMT labeling efficiency is checked before off-line separation. Finally, phosphopeptide intensities were adjusted based on total TMT reporter ion intensity in each channel to correct for slight mixing errors of each individual sample in the multiplex.

P35, lines 14 - please include details of how the off-line separation was performed of the TMT-labelled samples.

Our response:

The desalted multiplex was dried by vacuum centrifugation and separated by offline pentafluorophenyl (PFP)-based reversed phase HPLC fractionation as published (Grassetti et al., 2017). Briefly, TMT-labeled phosphopeptides were separated over a gradient of 5%-55% Buffer B from 0 to 61 mins. Forty-eight fractions were collected and concatenated into 24 by mixing the n^{th} and $n^{\text{th}}+24^{\text{th}}$ fraction. Buffer B: 95% ACN/0.1% TFA; Buffer A: 3% ACN/0.1% TFA

This clarifying text was added to the methods section.

P35, line 18 - 8% is a relatively high starting MeCN concentration for peptide elution from C18. Do you think this may be biasing your cohort of identified phosphopeptide motifs given that you will likely not be seeing any of the really hydrophilic peptides? I would be interested to see how this changes from ~3% MeCN.

Our response:

Great catch by the reviewer! We apologize for this error in reporting our methods. The actual gradient for TMT-labeled phosphopeptides is as follows: after loading, from 0% to 3% B over 2 minutes; from 3% to 22% B over 95 minutes; from 22% to 37% B over 25 minutes, followed by washing at 95% B and re-equilibration at 0% B for 6 and 8 minutes, respectively, where buffer A: 3% MeCN/0.125% formic acid and buffer B: 95% MeCN/0.125% formic acid. We have corrected this error in the Methods section of the manuscript as well.

P35, line 22 - please state how much calyculin A (activity units) were used - what were the reaction conditions/buffer?

Our response:

Done.

Thank you for submitting a revised manuscript for our consideration. It has now been reviewed once more by all three original reviewers, who generally found the work substantially improved towards becoming acceptable for publication. However, while referees 1 and 3 only request minor final changes, you will see that referee 2 retains one major reservation regarding the analyses of ADAM17 phosphorylation sites and their functional/physiological significance - which in my opinion appears well-taken. I therefore feel that this concern should be addressed in an exceptional second round of revision, ideally with additional data.

Furthermore, there are also a number of editorial issues that should be addressed during this final round of revision.

REFEREE REPORTS

Referee #1:

The authors have improved the quality of the manuscript and have addressed my major and minor points.

My last comment is that I noticed a difference with the numbers 512 and 289 (Figure 2C and D) and the numbers 548 and 398 in the main text. I am not sure why all the phosphosites "significantly increased in phosphorylation upon B56 versus control inhibitor expression" are not represented in the IceLogo.

Referee #2:

The revised manuscript is generally improved.

Pertaining to my main comment regarding the physiologic importance of presumed changes in phosphorylation of ADAM17 itself in the context of the ADAM17 SLIM variants (both the one with increased as well as the one with decreased PP2A-B56 binding), I am, however, not completely satisfied with the authors' revisions.

I acknowledge that the authors have shown in various ways, i.e. in their initial screen + by directly showing the phosphorylation changes in the ADAM17 I761A, but (still) not the LEE mutant, that direct ADAM17 (de)phosphorylation is changed by manipulating its PP2A-B56-binding SLIM. This was e.g. further corroborated by an additional *in vitro* dephosphorylation experiment on a PKA-phosphorylated ADAM17 fragment (in which phosphorylation, and hence dephosphorylation, occurred on non-identified sites). The authors have indeed gone at length to show that manipulation of the SLIM results in changes in ADAM17 phosphorylation in at least two, if not three phosphorylation sites (T735, S791 and S808). However, when asked about potential complementation of the phenotype in their ADAM17 *-/-* cell lines using phospho-mimetic or non-phospho-mimetic mutants of these sites, they chose to perform these experiment with mutants in which only one of these sites is mutated. Of course, these mutants (T735A and T735D) behave as wild-type proteins in these experiments....

If authors put so much effort in convincing us about the modulation of at least three ADAM17 phosphorylation sites by mutation of the ADAM17 PP2A-B56-binding SLIM, why didn't they use then a triple-A or triple-D ADAM17 mutant in these complementation experiments?

Still, in the abstract and the discussion, it is claimed that 'dephosphorylation of ADAM17 decreases growth factor signaling and tumor development in mice', and 'the PP2A-B56 holoenzyme reverts ADAM17 phosphorylation to limit shedding activity', but none of these claims are, at this point, firmly sustained by the data. Only by using the triple phospho-site mutants as potential rescue or non-rescue constructs, the authors will be able to make such statements (or not...). At this point, one can only guess whether the phenotypes (in proliferation, shedding, tumor growth etc..) seen with the ADAM17 I761A mutant, or the ADAM17 LEE mutant, are indeed DUE TO corresponding changes in ADAM17 phosphorylation at the three identified sites.

Referee #3:

The authors have done a good job of addressing the comments/concerns of all the reviewers. I would request however, that they include the information regarding the normalisation of the TMT-data post phosphopeptide enrichment in the main manuscript (methods section) as described in the reviewers' response.

Referee #1:

The authors have improved the quality of the manuscript and have addressed my major and minor points.

My last comment is that I noticed a difference with the numbers 512 and 289 (Figure 2C and D) and the numbers 548 and 398 in the main text. I am not sure why all the phosphosites "significantly increased in phosphorylation upon B56 versus control inhibitor expression" are not represented in the IceLogo.

Our response:

The 548 and 398 refer to all significantly increased phosphorylation sites (single, double, triple) in the respective condition, while 512 and 289 are only single phosphorylation sites.

Referee #2:

The revised manuscript is generally improved.

Pertaining to my main comment regarding the physiologic importance of presumed changes in phosphorylation of ADAM17 itself in the context of the ADAM17 SLIM variants (both the one with increased as well as the one with decreased PP2A-B56 binding), I am, however, not completely satisfied with the authors' revisions. I acknowledge that the authors have shown in various ways, i.e. in their initial screen + by directly showing the phosphorylation changes in the ADAM17 I761A, but (still) not the LEE mutant, that direct ADAM17 (de)phosphorylation is changed by manipulating its PP2A-B56-binding SLIM. This was e.g. further corroborated by an additional in vitro dephosphorylation experiment on a PKA-phosphorylated ADAM17 fragment (in which phosphorylation, and hence dephosphorylation, occurred on non-identified sites). The authors have indeed gone at length to show that manipulation of the SLIM results in changes in ADAM17 phosphorylation in at least two, if not three phosphorylation sites (T735, S791 and S808). However, when asked about potential complementation of the phenotype in their ADAM17 *-/-* cell lines using phospho-mimetic or non-phospho-mimetic mutants of these sites, they chose to perform these experiment with mutants in which only one of these sites is mutated. Of course, these mutants (T735A and T735D) behave as wild-type proteins in these experiments....

If authors put so much effort in convincing us about the modulation of at least three ADAM17 phosphorylation sites by mutation of the ADAM17 PP2A-B56-binding SLIM, why didn't they use then a triple-A or triple-D ADAM17 mutant in these complementation experiments?

Still, in the abstract and the discussion, it is claimed that 'dephosphorylation of ADAM17 decreases growth factor signaling and tumor development in mice', and 'the PP2A-B56 holoenzyme reverts ADAM17 phosphorylation to limit shedding activity', but none of these claims are, at this point, firmly sustained by the data. Only by using the triple phospho-site mutants as potential rescue or non-rescue constructs, the authors will be able to make such statements (or not...). At this point, one can only guess whether the phenotypes (in proliferation, shedding, tumor growth etc..) seen with the ADAM17 I761A mutant, or the ADAM17 LEE mutant, are indeed DUE TO corresponding changes in ADAM17 phosphorylation at the three identified sites.

Our response:

We appreciate the points raised by the reviewer. However, we have to say that we disagree somewhat on these points. First, we do not claim that the three ADAM17 phosphorylation sites identified to be regulated by PP2A-B56 are the sole sites causing the phenotypes we observe. Rather, we actually carefully state on page 15-16: “We anticipate that PP2A-B56 when bound to ADAM17 might also regulate the phosphorylation status of binding partners such as iRhom1/2 and that this contributes to regulation of shedding.” Thus, as we see it, the three phosphorylation sites identified are just three examples out of potentially many sites (on ADAM17 itself and on ADAM17 interacting proteins) being regulated by PP2A-B56. ADAM17 can be activated in response to multiple different signaling cues, involving many different kinases and phosphorylation sites. To tease out which sites regulated by PP2A-B56 may be relevant in which signaling context we find is beyond the scope of this work.

However, we appreciate that the wording in three instances can be misunderstood to suggest we make too far-reaching claims as pointed out by the reviewer. We have below suggested changes to these instances to make it clearer and reflect our results accurately. In our view, these changes will address the main concern of the reviewer. We want to point out that ADAM17 T735 and S808 phosphorylations have been shown by others to regulate ADAM17 shedding activity under certain experimental conditions and we refer to these papers in the discussion.

In abstract:

From

Dephosphorylation of ADAM17 decreases growth factor signaling and tumor development in mice.

To

Binding of PP2A-B56 to ADAM17 decreases growth factor signaling and tumor development in mice.

In results:

From

Thus, PP2A-B56 regulates physiologically relevant phosphorylation sites on ADAM17 to modulate its shedding activity. We anticipate that PP2A-B56 when bound to ADAM17 might also regulate the phosphorylation status of binding partners such as iRhom1/2 and that this contributes to regulation of shedding.

To

Thus, PP2A-B56 regulates physiologically relevant phosphorylation sites on ADAM17. We anticipate that PP2A-B56 when bound to ADAM17 regulates several

phosphorylation sites in the C-terminal tail as well as the phosphorylation status of binding partners such as iRhom1/2 and that this collectively controls shedding.

From discussion:

From

Yet, how ADAM17 becomes deactivated is not clear. Here we have revealed a novel inhibitory mechanism, whereby the PP2A-B56 holoenzyme reverts ADAM17 phosphorylation to limit its shedding activity. We identified three PP2A-B56 regulated sites on ADAM17 (Thr735, Ser791 and Ser808), of which Thr735 and Ser808 have been shown to be phosphorylated in response to cellular stress and enhance ADAM17 mediated shedding of EGFR ligands (Xu & Derynck 2010; Prakasam et al. 2014).

To

Yet, how ADAM17 becomes deactivated is not clear. Here we have revealed a novel inhibitory mechanism, whereby the PP2A-B56 holoenzyme reverts ADAM17 phosphorylations. We identified three PP2A-B56 regulated sites on ADAM17 (Thr735, Ser791 and Ser808), of which Thr735 and Ser808 have been shown to be phosphorylated in response to cellular stress and enhance ADAM17 mediated shedding of EGFR ligands (Xu & Derynck 2010; Prakasam et al. 2014).

Referee #3:

The authors have done a good job of addressing the comments/concerns of all the reviewers. I would request however, that they include the information regarding the normalisation of the TMT-data post phosphopeptide enrichment in the main manuscript (methods section) as described in the reviewers' response.

Done.

Thank you for submitting your final revised manuscript for our consideration. I am pleased to inform you that we have now accepted it for publication in The EMBO Journal.

Corresponding Author Name: Jakob Nilsson

Manuscript Number: EMBOJ-2019-103695